# D&R: Recovery-based AI-Generated Text Detection via a Single Black-box LLM Call

**Yuxia Sun[1], Ran Zhang[1], Aoxiang Sun[2], Xu Li[1], Zitao Liu[3], Jingcai Guo[4]***

[1]College of Information Science and Technology, Jinan University, Guangzhou, China
[2]College of Information and Computational Science, Jilin University, China
[3]Guangdong Institute of Smart Education, Jinan University, Guangzhou, China
[4]Department of Computing/LSGI, The Hong Kong Polytechnic University, Hong Kong SAR
`tyxsun@email.jnu.edu.cn, wwren@stu2024.jnu.edu.cn,`
`sunax24@mails.jlu.edu.cn, lixu1024@stu2024.jnu.edu.cn,`
`liuzitao@jnu.edu.cn, jc-jingcai.guo@polyu.edu.hk`

## Abstract

Large language models (LLMs) generate increasingly human-like text, raising concerns about misinformation and authenticity. Detecting AI-generated text remains challenging: existing methods often underperform, especially on short texts, require probability access unavailable in real-world black-box settings, incur high costs from multiple calls, or fail to generalize across models. We propose Disrupt-and-Recover (D&R), a recovery-based detection framework grounded in posterior concentration. D&R disrupts text via model-free Within-Chunk Shuffling, performs a single black-box LLM recovery, and measures semantic–structural recovery similarity as a proxy for concentration. This design ensures efficiency, black-box practicality, and is theoretically supported under the concentration assumption. Extensive experiments across four datasets and six source models show that D&R achieves state-of-the-art performance, with AUROC 0.96 on long texts and 0.87 on short texts, surpassing the strongest baseline by +0.08 and +0.14. D&R further remains robust under source–recovery mismatch and model variation. Our code and data is available at `https://github.com/Yuxia-Sun/D-R`.

## 1 Introduction

Large language models (LLMs) have rapidly advanced to generate human-like text across domains such as education, news, scientific writing, and online communication. While these advances create tremendous opportunities, they also raise serious concerns about misinformation, academic integrity, and content authenticity, making reliable AI-generated text detection increasingly crucial. However, this task remains highly challenging. Real-world applications require detectors that can efficiently scale to large volumes of text with minimal overhead, for example by reducing LLM calls. They must remain robust to evolving and diverse source models while operating in black-box settings without probability access. They must also handle varied text lengths, with short texts being particularly difficult. These challenges underscore the need for a detection framework that is not only accurate but also efficient, black-box practical, generalizable, and robust.

Despite recent progress, the performance of existing AI-text detectors remains far from satisfactory, even on common long-text settings. Likelihood- and entropy-based methods (Gehrmann et al., 2019; Hashimoto et al., 2019) rely on white-box access to model probabilities, making them impractical for black-box settings. Perturbation- and continuation-based methods (Bao et al., 2024; Yang et al., 2024) may improve accuracy, and rewriting-based methods (Mao et al., 2024; Park et al., 2025) avoid probability access, but all require multiple model calls, incurring high computational cost and showing instability (particularly on short texts). Supervised classifiers (OpenAI, 2019) lack generalization and require costly labels, while watermarking detectors (Zhao et al., 2024) heavily depend

---

*Jingcai Guo is the corresponding author.

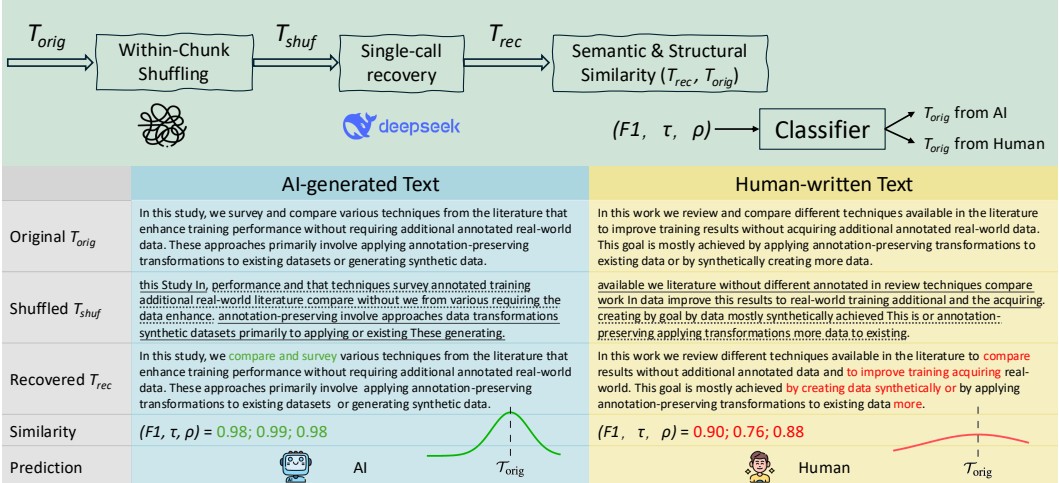

Figure 1: Illustration of D&R. **Top:** pipeline overview. **Bottom:** AI vs. Human examples. In the Shuffled $\mathcal{T}_{\text{shuf}}$, individual chunks are highlighted with alternating solid and dashed underlines. AI text is recovered with fewer errors/higher similarity (green), whereas Human text shows more errors/lower similarity (red). The curves below each column schematically show recovery outcomes across examples: recoveries from AI-generated text concentrate more sharply around $\mathcal{T}_{\text{orig}}$, while those from human-written text are more dispersed.

on model providers. Consequently, no existing method simultaneously delivers high performance while satisfying the demands of efficiency, black-box practicality, generalizability, and robustness.

To address these limitations, we propose *Disrupt-and-Recover (D&R)*, a recovery-based detection framework grounded in the observation of posterior concentration: when text is disrupted in a way consistent with LLM pretraining biases, AI-generated text yields LLM-based recoveries that concentrate more sharply around the original text $\mathcal{T}_{\text{orig}}$, whereas human-written text produces more dispersed recoveries. D&R follows three key steps: (i) apply a model-free disruption, *Within-Chunk Shuffling (WCS)*, which aligns with pretraining objectives and constrains recovery to a reduced candidate space; (ii) perform a single black-box recovery call, avoiding the inefficiency of multi-call methods; and (iii) compute semantic and structural recovery similarities between the recovered and original text, which serve as observable proxies for posterior concentration. These similarities are then passed to a lightweight binary classifier to output the final prediction. As illustrated in Figure 1, AI text (left) recovers with higher similarity and more concentrated outcomes near $\mathcal{T}_{\text{orig}}$, while Human text (right) shows lower similarity and more dispersed recoveries. Building on this concentration assumption, we establish the theoretical foundation of D&R. We then validate its effectiveness, robustness, and usability through extensive experiments across diverse datasets, source models, and scenarios.

This paper makes the following four contributions:

- We introduce D&R, a novel detection framework that employs a model-free disruption, Within-Chunk Shuffling (WCS), conceptually aligned with the inductive biases of LLM pretraining, and ensures efficiency through a single black-box recovery call.

- We propose the Concentration Assumption and design semantic–structural recovery similarity metrics as faithful proxies for posterior concentration, providing the theoretical rationale for D&R.

- We conduct extensive experiments against eleven representative baselines across diverse datasets and source models, showing that D&R achieves state-of-the-art detection and consistently surpasses the strongest baseline (RAIDAR) by a significant margin.

- We further show that D&R remains robust and practical in challenging scenarios, including short texts, source–recovery model mismatch, and recovery-model variation.

## 2 RELATED WORK

**Zero-shot detectors.** Zero-shot detectors avoid large labeled datasets and instead exploit unsupervised signals such as probabilities or perturbations. Likelihood- and entropy-based methods (Hashimoto et al., 2019; Gehrmann et al., 2019) depend on model probability distributions, making them inherently white-box detectors. To address these limitations, recent works propose estimating black-box logits via proxy model tuning (Zeng et al., 2024), utilizing dual-model scoring (Hans et al., 2024), or analyzing intrinsic features by pre-trained models (Yu et al., 2024). Perturbation-based approaches such as DetectGPT (Mitchell et al., 2023) analyze the log-likelihood curvature of perturbed passages, with the state-of-the-art variant Fast-DetectGPT (Bao et al., 2024) improving both accuracy and efficiency. NPR (Su et al., 2023) also leverages paraphrasing but measures residual signals, and is less effective than Fast-DetectGPT. Continuation-based methods such as DNA-GPT (Yang et al., 2024) truncate the text and regenerate the suffix for comparison. Rewriting-based RAIDAR (Mao et al., 2024) generates paraphrased versions of entire passages and measures consistency via edit distances between versions, but it requires multiple model calls, depends on a specific paraphraser, and is vulnerable to prompt-level manipulations. Among these, RAIDAR is most relevant to our D&R, as both rely on transformation-consistency—assessing textual consistency under transformations such as paraphrasing or shuffling–recovery. In contrast to existing generative methods (e.g., perturbation-, continuation-, and rewriting-based ones), D&R achieves detection with only a single model call.

**Non-zero-shot detectors.** Non-zero-shot detectors rely on supervised discriminative models trained with large labeled datasets. Representative examples include RoBERTa-based classifiers (Liu et al., 2019), and the OpenAI Text Classifier (OpenAI, 2019), alongside recent frameworks utilizing multi-level contrastive learning (Guo et al., 2024), stylistic alignment (Chen et al., 2025), or out-of-distribution detection on human texts (Zeng et al., 2025). These approaches can achieve strong in-domain accuracy but generalize poorly to unseen generation models and require costly labeling and frequent retraining as LLMs evolve. Another line of work explores watermarking (Zhao et al., 2024; Kirchenbauer et al., 2023), which embeds detectable signatures during generation. However, watermarking depends on model providers and is unsuitable for post-hoc detection.

## 3 METHOD

Our D&R method follows the pipeline shown in Algorithm 1. First, we introduce a semantic-preserving disruption, Within-Chunk Shuffling, which aligns with LLM pretraining objectives by constraining the recovery problem to a locally permuted candidate space. Second, we perform a single LLM call to recover the text. Next, we compute both semantic and structural similarities between the recovered and source texts, and use these similarities as observable recoverability metrics. Finally, we train a binary classifier on the recoverability metrics of labeled AI-generated and human-written texts, and apply it to obtain detection results.

---

**Algorithm 1** D&R Pipeline

---

1: **Input:** Original text $\mathcal{T}_{\text{orig}}$; black-box LLM $\mathcal{M}$
2: **Output:** Prediction $y \in \{\text{Human, AI}\}$
3: **for** each chunk $c_i$ in $\mathcal{T}_{\text{orig}}$ **do**
4:      $c_i^{\text{shuf}} \leftarrow \text{ShuffleTokens}(c_i)$            ▷ Apply Within-Chunk Shuffling
5: **end for**
6: $\mathcal{T}_{\text{shuf}} \leftarrow \text{Join}(\{c_i^{\text{shuf}}\})$            ▷ Obtain disruption result
7: $\mathcal{T}_{\text{rec}} \leftarrow \mathcal{M}.\text{Recover}(\mathcal{T}_{\text{shuf}})$            ▷ Single-call recovery
8: $F_1 \leftarrow \text{SemanticSim}(\mathcal{T}_{\text{orig}}, \mathcal{T}_{\text{rec}})$            ▷ Compute semantic similarity
9: $(\tau, \rho) \leftarrow \text{StructuralSim}(\mathcal{T}_{\text{orig}}, \mathcal{T}_{\text{rec}})$            ▷ Compute structural similarity
10: $y \leftarrow \text{Classifier}([F_1, \tau, \rho])$            ▷ Predict label
11: **return** $y$

---

Intuitively, recovery outcomes for AI-generated text within this constrained space tend to be highly concentrated, whereas human-written text yields more dispersed results due to the diversity of writing processes. This concentration gap can be characterized by the notion of *posterior concentration*,

which provides the theoretical rationale behind our method and is, in practice, approximated by recovery similarity metrics.

A key design in D&R is the disruption step, which determines the nature of the subsequent recovery task. We adopt *Within-Chunk Shuffling* (WCS), where the original text $\mathcal{T}_{\text{orig}}$ is segmented into chunks by punctuation marks, and tokens within each chunk are randomly permuted while preserving chunk order. This disruption requires no model calls and can be implemented with a simple random shuffling function.

The advantage of WCS is that it constrains recovery to a locally permuted candidate space rather than the unconstrained generative space, closely aligning with pretraining objectives that emphasize predicting local token orderings. As a result, recovery under WCS becomes almost effortless for the LLM, akin to recalling the original token order, leading to recovered texts that lie very close to the source. In distributional terms, AI-generated text tends to yield recovery outcomes that are highly concentrated near the original text, i.e., exhibiting strong *posterior concentration*, whereas human-written text produces more dispersed recoveries due to greater variability in writing processes.

Formally, as Algorithm 1 shows, we segment $\mathcal{T}_{\text{orig}}$ into chunks by punctuation, apply a random permutation to the tokens within each chunk, and then join the shuffled chunks to obtain the disrupted text $\mathcal{T}_{\text{shuf}}$, which is used as the input for the subsequent recovery step.

## 3.1 RECOVERY WITH A SINGLE LLM CALL

After disruption, the shuffled text $\mathcal{T}_{\text{shuf}}$ is passed to a large language model for recovery. We perform this step with a *single* LLM call, where the model is prompted to restore token order and reconstruct a coherent version of the original text $\mathcal{T}_{\text{rec}}$. A typical recovery prompt is:

```
The following text has its tokens shuffled within
punctuation-delimited spans.  Please restore the correct
word order without adding or removing words:  [INPUT].
```

This single-call design is both more efficient than multi-call approaches and well aligned with LLM pretraining priors, as predicting local token order is a task for which pretrained models are already highly competent. In practice, recovery can be performed either (i) via API calls to black-box LLMs (e.g., `DeepSeek-v3`), or (ii) via local inference with smaller models (e.g., `Mistral 7B`). Importantly, D&R achieves strong performance in both settings, demonstrating robustness and cost-effectiveness, a property we further validate in our Recovery-Model Independence experiments (see Section 4.3). Formally, given disrupted input $\mathcal{T}_{\text{shuf}}$ and recovery model $\mathcal{M}$, the recovered text is obtained as in Algorithm 1.

## 3.2 RECOVERABILITY METRICS

Given an original text $\mathcal{T}_{\text{orig}}$ and its recovered counterpart $\mathcal{T}_{\text{rec}}$, we quantify *recoverability* using two complementary forms of recovery similarity: semantic and structural.

**Semantic similarity.** We adopt *BERTScore* (Zhang et al., 2020), which measures token-level semantic overlap by comparing contextual embeddings from a pre-trained transformer (*bert-base-uncased*, Devlin et al., 2019). Let $m$ and $n$ denote the number of tokens in $\mathcal{T}_{\text{orig}}$ and $\mathcal{T}_{\text{rec}}$, respectively; $\{x_i\}_{i=1}^{m}$ and $\{y_j\}_{j=1}^{n}$ their contextual embeddings; and $\cos(\cdot, \cdot)$ the cosine similarity. Semantic similarity is defined by the F1 score calculated from BScore's precision and recall:

$$\text{Precision (P)} = \frac{1}{n} \sum_{j=1}^{n} \max_{i} \cos(x_i, y_j), \quad \text{Recall (R)} = \frac{1}{m} \sum_{i=1}^{m} \max_{j} \cos(x_i, y_j), \quad \text{F1} = \frac{2PR}{P + R}.$$

**Structural similarity.** We measure word-order consistency using Kendall's $\tau$ (Kendall, 1938; Chen et al., 2023) and Spearman's $\rho$ (Spearman, 1904; Guo et al., 2025), two rank-based correlation coefficients applied to token orderings. When $m \neq n$ or tokens repeat, we first construct a one-to-one alignment $A = \{(i_k, j_k)\}_{k=1}^{\ell} \subseteq [m] \times [n]$ (e.g., via token-normalized Longest Common Subsequence (LCS) with left-to-right stable matching), and then compute ranks $r_k = i_k$ and $s_k = j_k$ for $k = 1, \ldots, \ell$, where $C$ and $D$ denote the numbers of concordant and discordant pairs among the

aligned indices.

$$\tau = \frac{C - D}{\frac{1}{2}\,\ell(\ell-1)}, \qquad \rho = 1 - \frac{6\sum_{k=1}^{\ell}(r_k - s_k)^2}{\ell(\ell^2 - 1)}.$$

Semantic similarity captures fidelity of meaning, while structural similarity assesses reconstruction of word order. Together, higher values indicate the recovered text stays closer to the source both semantically and structurally. Thus, these complementary metrics provide observable signals of recoverability.

**Sanity check 1.** We conducted a lightweight evaluation of our metrics on the ML-ArXiv-Papers dataset (arXiv.org submitters, 2024) by sampling 1,000 AI-generated and 1,000 human-written texts. For each text, we computed recovery similarity with three metrics (BERTScore F1, Kendall's $\tau$, and Spearman's $\rho$) and plotted their distributions in Figure 2. AI-generated texts consistently exhibit higher average similarity scores than human-written texts, revealing clear distributional gaps across all metrics. Thus, *our recoverability metrics are effective for distinguishing AI-generated from human-written text.*

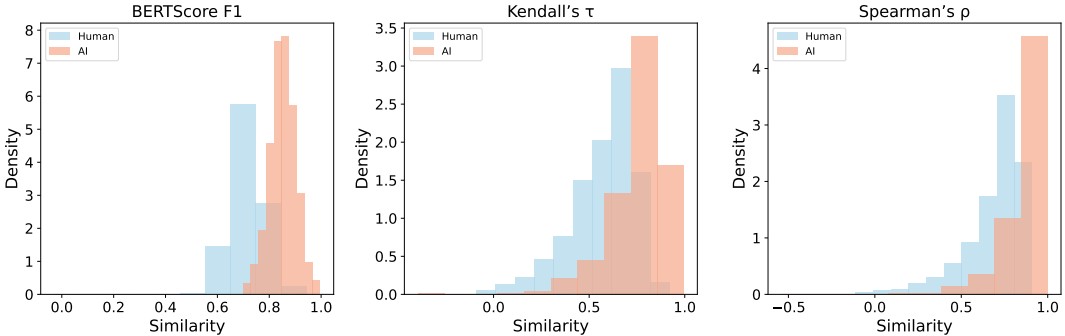

Figure 2: Distributions of recovery similarity scores on 1,000 ML-ArXiv-Papers samples. AI-generated texts show a clear trend toward higher scores than human-written texts across all metrics: BERTScore F1, Kendall's $\tau$, and Spearman's $\rho$.

**Sanity check 2.** We conducted another lightweight evaluation on the ML-ArXiv-Papers dataset (arXiv.org submitters, 2024), varying the temperature of the recovery model to control output concentration, where the temperature is a decoding hyperparameter in the inference API (lower values yield more concentrated outputs, while higher values yield more dispersed ones). As shown in Figure 3, recovery similarity scores across all metrics decrease as temperature increases, demonstrating a positive correlation with posterior concentration. This confirms that *our recoverability metrics provide observable proxies for posterior concentration.*

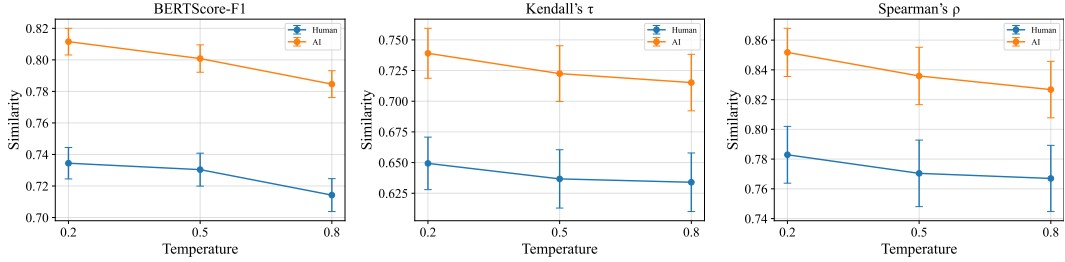

Figure 3: Recovery similarity scores on 1,000 ML-ArXiv-Papers samples across recovery-model temperature settings. Scores track posterior concentration: lower temperatures yield higher concentration and similarity, whereas higher temperatures reduce both across all metrics.

### 3.3 THEORETICAL ANALYSIS AND PROOF

#### 3.3.1 THEORETICAL RATIONALE

**Concentration Assumption:** Following a disruption that preserves semantics while respecting the inductive biases of LLM pretraining (e.g., Within-Chunk Shuffling), the distribution of LLM-based recovery outputs for AI-generated text is more concentrated in the vicinity of the original text, whereas the distribution for human-written text is more dispersed.

Crucially, since posterior concentration is a distributional property that cannot be directly observed in a single-call recovery, D&R *indirectly estimates* it from a *single recovery sample* $\mathcal{T}_{\text{rec}}$ by computing its similarity $S$ to the original text $\mathcal{T}_{\text{orig}}$. We prove below that **Recovery similarity is a faithful proxy for posterior concentration**, yielding a non-trivial gap between AI- and human-written texts.

SETUP

Let $\mathcal{M}$ be the recovery model, $\mathcal{T}_{\text{orig}}$ the original text, $\mathcal{T}_{\text{shuf}}$ its WCS-disrupted version, and $\mathcal{T}_{\text{rec}} \sim \mathcal{M}(\cdot \mid \mathcal{T}_{\text{shuf}})$ a recovered sample. Define a distance $d(\cdot, \cdot) \geq 0$ (e.g., normalized Kendall distance) and a bounded similarity $S(\cdot, \cdot) \in [0, 1]$ with $S = 1$ when texts match.

The posterior is $(r, \delta)$-*concentrated* if $\Pr\big(d(\mathcal{T}_{\text{orig}}, \mathcal{T}_{\text{rec}}) \leq r\big) \geq 1 - \delta$. Assume $S$ is continuous in $d$ with modulus of continuity $\omega(\cdot)$, i.e., $S(\mathcal{T}_{\text{orig}}, u) \geq 1 - \omega(d(\mathcal{T}_{\text{orig}}, u))$, $\forall u$, with $\omega(0) = 0$ and $\omega$ non-decreasing (e.g., $\omega(t) = Lt$).

**Theorem 1 (Posterior concentration $\Rightarrow$ high recovery similarity).** If the recovery posterior is $(r, \delta)$-concentrated, then with probability at least $1 - \delta$, $S(\mathcal{T}_{\text{orig}}, \mathcal{T}_{\text{rec}}) \geq 1 - \omega(r)$, and consequently,

$$\mathbb{E}\big[S(\mathcal{T}_{\text{orig}}, \mathcal{T}_{\text{rec}})\big] \geq (1 - \delta)(1 - \omega(r)).$$

*Proof.* Define $A = \{\mathcal{T}_{\text{rec}} : d(\mathcal{T}_{\text{orig}}, \mathcal{T}_{\text{rec}}) \leq r\}$. By posterior concentration, $\Pr(A) \geq 1 - \delta$. For $\mathcal{T}_{\text{rec}} \in A$, the continuity of $S$ gives $S \geq 1 - \omega(r)$. For $\mathcal{T}_{\text{rec}} \notin A$, we only know $S \geq 0$. Hence $\mathbb{E}[S] \geq (1 - \omega(r)) \Pr(A) \geq (1 - \omega(r))(1 - \delta)$. $\square$

**Theorem 2 (Non-trivial gap under Concentration Assumption).** Let $\mathcal{T}_{\text{orig}}^{\text{AI}}$ and $\mathcal{T}_{\text{orig}}^{\text{Human}}$ denote AI-generated and human-written texts. Suppose their recovery posteriors are $(r_A, \delta_A)$- and $(r_H, \delta_H)$-concentrated, respectively. Assume there exists $\delta_0 > 0$ and $\epsilon > 0$ such that the expected similarity for human text satisfies: $\mathbb{E}[S(\mathcal{T}_{\text{orig}}^{\text{Human}}, \mathcal{T}_{\text{rec}}^{\text{Human}})] < (1 - \delta_0)(1 - \omega(r_H))$, and $(1 - \delta_A)(1 - \omega(r_A)) \geq (1 - \delta_H)(1 - \omega(r_H)) + 2\epsilon$. Furthermore, assume the compatibility condition $\delta_H \geq \delta_0 \geq \delta_H - \frac{\epsilon}{1 - \omega(r_H)}$ holds and $\omega(r_A) \leq \omega(r_H)$. Then

$$\mathbb{E}[S(\mathcal{T}_{\text{orig}}^{\text{AI}}, \mathcal{T}_{\text{rec}}^{\text{AI}})] \geq \mathbb{E}[S(\mathcal{T}_{\text{orig}}^{\text{Human}}, \mathcal{T}_{\text{rec}}^{\text{Human}})] + \epsilon.$$

*Proof.* By Theorem 1, $\mathbb{E}[S_{\text{AI}}] \geq (1 - \delta_A)(1 - \omega(r_A))$; by the assumption, $\mathbb{E}[S_{\text{Human}}] < (1 - \delta_0)(1 - \omega(r_H))$. Therefore, $\mathbb{E}[S_{\text{AI}}] - \mathbb{E}[S_{\text{Human}}] \geq (1 - \delta_H)(1 - \omega(r_H)) + 2\epsilon - (1 - \delta_0)(1 - \omega(r_H)) = (1 - \omega(r_H))(\delta_0 - \delta_H) + 2\epsilon \geq \epsilon$. $\square$

*Consequences for Metrics.* - *Kendall $\tau$:* $\tau = 1 - 2d$, so $\omega(r) = 2r$. - *Spearman $\rho$:* If at most fraction $r$ of ranks are perturbed, then $1 - \rho \leq c_\ell r$, hence $\omega(r) = c_\ell r$. - *BERTScore F1:* WCS preserves token sets; embedding drift under local permutations is bounded by $L_{\text{sem}} r$, hence $\omega(r) = L_{\text{sem}} r$.

**Takeaway.** Theorem 1 shows that posterior concentration entails high recovery similarity: as $r \to 0$ and $\delta \to 0$, $S \to 1$. Theorem 2 shows that under the Concentration Assumption, AI texts achieve strictly higher expected recovery similarity than human texts by margin $\epsilon$. Thus recovery similarity is a faithful proxy for posterior concentration, providing the theoretical foundation for D&R. We provide a detailed discussion on the validity of the assumptions underlying Theorem 2 in **Appendix A.2**.

#### 3.3.2 COMPUTATIONAL OVERHEAD.

The efficiency of D&R stems from requiring only a single black-box LLM call. Its time overhead is $T_{\text{D\&R}} = T_{\text{shuffle}} + T_{\text{LLM}} + T_{\text{similarity}}$, where the shuffling cost is negligible ($T_{\text{shuffle}} \approx 0$) and the similarity scoring cost is much smaller than an LLM call ($T_{\text{similarity}} \ll T_{\text{LLM}}$), so the overall cost is

dominated by one call. In contrast, existing generative methods (e.g., perturbation-, continuation-, or rewriting-based) require multiple calls, performing $k > 1$ queries with overhead $T_{baseline} \approx k \cdot T_{LLM} + T_{extra}$, which scales as $O(k \cdot T_{LLM})$. Thus, D&R lowers detection overhead to $O(T_{LLM})$, providing linear efficiency gains without additional assumptions. Empirical validation of these efficiency gains is provided in Appendix A.4.

## 4 EXPERIMENTS

We evaluate D&R against representative baselines on long-text datasets, with ablations of recoverability metrics, and further analyze its usability and robustness in challenging real-world scenarios including short texts, source–recovery model mismatch, and recovery LLM variation, providing a comprehensive assessment of its effectiveness.

### 4.1 SETTINGS

**Datasets and Metrics.** To evaluate the performance of the D&R method on paragraph-level AI-generated text detection, we use six publicly available datasets spanning different text lengths and domains. Based on average text length, we group them into long texts ($>800$ words) and short texts ($<350$ words), with dataset-wise length distributions shown in Figure 5. The long-text group includes ML-ArXiv-Papers (research abstracts) (arXiv.org submitters, 2024), CNN-DailyMail (news articles) (See et al., 2017), IMDB (movie reviews) (Maas et al., 2011), and ROCStories (five-sentence stories) (Mostafazadeh et al., 2016); the short-text group includes Wikihow (instructional guides) (Sentence-Transformers, 2020), AG-News (news headlines and summaries) (Zhang et al., 2015), and Reddit (user-generated posts) (Sentence-Transformers, 2021). For each dataset, we sample 1,000 human-written texts as negatives and generate AI counterparts of comparable length via paraphrasing prompts from different source models, ensuring balanced parallel datasets. To mimic diverse real-world source models, we use six widely adopted LLMs from different providers: GPT-2 (Solaiman et al., 2019), GPT-Neo-2.7B (Black et al., 2021), Qwen-Turbo (Yang et al., 2025), GPT-4.1 (OpenAI, 2025), Gemini2.5-Flash (Comanici et al., 2025), and Grok3 (xAI, 2025). Detection performance is measured with the area under the ROC curve (AUROC) (Fawcett, 2006), which reflects the probability that the detector ranks an AI-generated text above a human-written one.

**Baselines.** We compare D&R against eleven representative baselines spanning all major families of AI-generated text detection. These include the rewriting-based RAIDAR (Mao et al., 2024), most closely related to our method; the perturbation-based Fast-DetectGPT (Bao et al., 2024), the state of the art in this family; the continuation-based DNA-GPT (Yang et al., 2024); the likelihood-based method (Hashimoto et al., 2019); recent works such as Binoculars (Hans et al., 2024), DALD (Zeng et al., 2024), and Text Fluoroscopy (Yu et al., 2024). Finally, we compare against supervised and learning-based detectors, including the RoBERTa classifier (OpenAI, 2019), Detective (Guo et al., 2024) , Imitate Before Detect (Chen et al., 2025), and Human-Outlier OOD detection (Zeng et al., 2025). In the main experiments, we report results against all eleven baselines for comprehensive coverage. For additional experimental analyses, we focus on the strongest baselines and report results only against RAIDAR and Fast-DetectGPT. Baseline configurations follow their original papers, unless otherwise specified.

### 4.2 MAIN RESULTS

**Detection Performance.** We compare D&R with eleven representative baselines on four long-text datasets across six source models, reporting AUROC performance in Table 1 and their visualization in Figure 4. For fairness, the same transformation model (DeepSeek-v3) is used for RAIDAR's paraphrasing and D&R's recovery. As shown in Table 1, D&R achieves the highest mean AUROC with the lowest variance ($0.9602 \pm 0.0351$), substantially outperforming all baselines in both accuracy and stability. Figure 4 further illustrates that D&R's advantage holds consistently, while baselines not only perform worse but also fluctuate with dataset and source model shifts. For instance, on ML-ArXiv-Papers, when the source model changes from GPT-2 to the more advanced Grok-3, RAIDAR drops from about 0.90 to 0.77, whereas D&R remains stable above 0.95. These results demonstrate that D&R is a robust and effective zero-shot detector for long-text scenarios.

Table 1: Mean±SD AUROC on four long-text datasets, averaged over six source models, using DeepSeek-v3 as the recovery model. The first two entries are traditional methods, while the remaining baselines represent recent state-of-the-art approaches from top venues (NeurIPS, ICML, ACL, EMNLP, ICLR), followed by our proposed D&R. Detailed per-dataset results are provided in Appendix A.5.

| Dataset | RoBERTa-based | Likelihood | Dald | OOD-based |
|---|---|---|---|---|
| ML-ArXiv-Papers | 0.6195±0.1443 | 0.6628±0.1667 | 0.7838±0.1441 | 0.7648±0.1323 |
| CNN-DailyMail | 0.6174±0.1456 | 0.5786±0.1186 | 0.7336±0.0992 | 0.6203±0.1703 |
| IMDB | 0.6114±0.1273 | 0.6617±0.1085 | 0.6941±0.1303 | 0.8392±0.0773 |
| ROCStories | 0.7675±0.1182 | 0.5852±0.2158 | 0.6084±0.2515 | 0.9109±0.0694 |
| *Avg.* | 0.6539±0.1498 | 0.6221±0.1633 | 0.7050±0.1313 | 0.7838±0.1373 |

| Dataset | ImBD | Text Fluoroscopy | DeTeCtive | Binoculars |
|---|---|---|---|---|
| ML-ArXiv-Papers | 0.7693±0.0964 | 0.8266±0.1367 | 0.7772±0.1123 | 0.6435±0.1406 |
| CNN-DailyMail | 0.8115±0.1248 | 0.8905±0.0762 | 0.8295±0.1116 | 0.5333±0.1044 |
| IMDB | 0.8424±0.0957 | 0.8917±0.0621 | 0.8452±0.0403 | 0.6650±0.0910 |
| ROCStories | 0.7185±0.1451 | 0.7399±0.1053 | 0.8756±0.0917 | 0.5054±0.1487 |
| *Avg.* | 0.7854±0.0905 | 0.8372±0.0951 | 0.8319±0.0890 | 0.5868±0.0962 |

| Dataset | DNA-GPT | Fast-DetectGPT | RAIDAR | **D&R***(ours)* |
|---|---|---|---|---|
| ML-ArXiv-Papers | 0.6400±0.1708 | 0.7242±0.1456 | 0.8611±0.0472 | **0.9266±0.0354** |
| CNN-DailyMail | 0.5953±0.1659 | 0.5838±0.1556 | 0.8471±0.0759 | **0.9830±0.0063** |
| IMDB | 0.6491±0.1291 | 0.7277±0.1075 | 0.8675±0.0552 | **0.9451±0.0314** |
| ROCStories | 0.6231±0.2002 | 0.6385±0.1855 | 0.9323±0.0482 | **0.9861±0.0115** |
| *Avg.* | 0.6269±0.1697 | 0.6685±0.1583 | 0.8770±0.0657 | **0.9602±0.0351** |

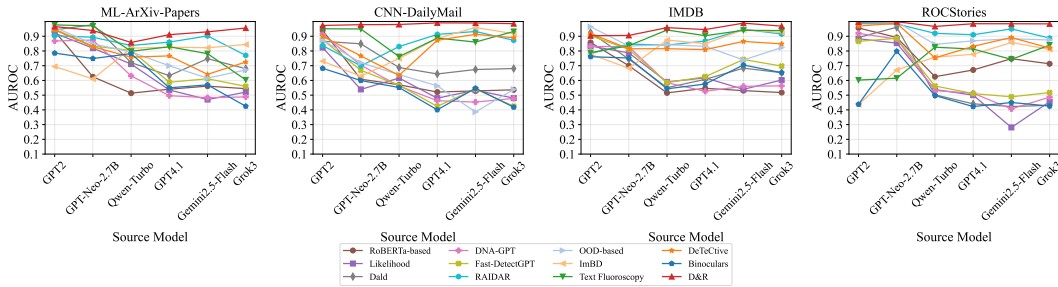

Figure 4: AUROC scatter plot on four long-text datasets across six source models, complementing the averaged results in Table 1.

**Ablation Study.** We examine the contribution of semantic and structural recovery similarities through an ablation study on four long-text datasets with advanced source models. As shown in Table 7, removing semantic similarity results in the largest performance drop (↓28.1%), while removing structural similarity also yields a substantial decrease (↓19.8%). The full model achieves an AUROC of 0.9614, demonstrating that both forms of recovery similarity are indispensable and that their combination ensures state-of-the-art accuracy and stability. We also experimentally showed that our Within-Chunk Shuffling (WCS) is optimal compared to global or chunk-order shuffling, effectively striking an optimal balance in the recovery task difficulty to maximize the concentration gap, detailed analyses for these experiments are provided in Appendix A.3.2.

## 4.3 ANALYSIS

**Short-text Robustness.** Short texts are particularly challenging for AI-generated text detection, as the limited context amplifies the distributional overlap between human and machine outputs. As shown in Table 2, D&R achieves the highest mean AUROC with low variance (0.8687±0.0888), significantly outperforming RAIDAR and Fast-DetectGPT by margins of 0.14 and 0.21, respectively. The advantage is most pronounced on earlier source models (GPT-2, GPT-Neo-2.7B), where D&R attains near-perfect AUROC scores (around 0.99), while on stronger models performance declines

for all methods but D&R still maintains clear margins. These results underscore D&R's consistent superiority on short-text detection and its resilience across both weaker and stronger generators.

Table 2: AUROC performance on three Short-Text datasets across six source models. For each dataset, results from two earlier models (GPT-2, GPT-Neo-2.7B) and four more advanced models (Qwen-Turbo, GPT-4.1, Gemini 2.5, Grok-3) are separated by a dotted line.

| Dataset | Source Model | Method | | |
|---|---|---|---|---|
| | | Fast-DetectGPT | RAIDAR | **D&R** (*ours*) |
| Wikihow | GPT2 | 0.7449 | 0.7800 | **0.9904** |
| | GPT-Neo-2.7B | 0.7936 | 0.7743 | **0.9987** |
| | Qwen-Turbo | 0.5193 | 0.5300 | **0.7363** |
| | GPT4.1 | 0.4551 | 0.5700 | **0.7850** |
| | Gemini2.5-Flash | 0.4370 | 0.7550 | **0.8517** |
| | Grok3 | 0.4719 | 0.6950 | **0.7727** |
| AG-News | GPT2 | 0.7780 | 0.7231 | **0.9886** |
| | GPT-Neo-2.7B | 0.7932 | 0.7524 | **0.9982** |
| | Qwen-Turbo | 0.7542 | 0.6850 | **0.7666** |
| | GPT4.1 | 0.5819 | 0.6735 | **0.8202** |
| | Gemini2.5-Flash | 0.6898 | 0.7776 | **0.8835** |
| | Grok3 | 0.6439 | 0.7375 | **0.7963** |
| Reddit | GPT2 | 0.7043 | 0.7649 | **0.9271** |
| | GPT-Neo-2.7B | 0.7259 | 0.7947 | **0.9586** |
| | Qwen-Turbo | 0.6852 | 0.7310 | **0.7502** |
| | GPT4.1 | 0.6406 | 0.7429 | **0.8451** |
| | Gemini2.5-Flash | 0.7007 | 0.7810 | **0.9007** |
| | Grok3 | 0.6916 | 0.7800 | **0.8672** |
| | Mean±SD | 0.6561±0.1129 | 0.7248±0.0707 | **0.8687±0.0888** |

**Source-model Agnosticism (Robustness under Model Mismatch).** As transformation-consistency based detectors, both D&R and RAIDAR rely on a transformation model to recover or paraphrase the text generated by a source model. Although neither method requires explicit knowledge of the source model, performance can depend on whether the source and transformation models are the same. We therefore evaluate two cases: (i) the same-source case (*src=trx*), an easier, pseudo–white-box setting in which the detector can implicitly benefit from the source model's distributional biases; and (ii) the different-source case (*src≠trx*), a more realistic heterogeneous pairing. As shown in Table 3, across both settings, our D&R consistently outperforms RAIDAR. Moreover, under the different-source condition, D&R degrades only 0.1-3.3% degradation (mean 1.9%), whereas RAIDAR drops by 4.2-14.2% (mean 9.4%). These results demonstrate that D&R is source-agnostic: it does not rely on knowledge of the source model, remaining markedly more robust than RAIDAR under model mismatch.

Table 3: AUROC performance under same vs. different Source–Transformation Pairings. The transformation model (*trx*) is fixed as DeepSeek-v3. For the 'Same' case (*src=trx*), the source model equals the transformation model; for the 'Different' case (*src≠trx*), results are averaged over six diverse source models listed in Table 1.

| Dataset | RAIDAR | | | **D&R** (*ours*) | | |
|---|---|---|---|---|---|---|
| | Same (*src=trx*) | Different (*src≠trx*) | | Same (*src=trx*) | Different (*src≠trx*) | |
| ML-ArXiv-Papers | 0.9475 | 0.8611 | ↓9.1% | 0.9590 | 0.9266 | ↓**3.3%** |
| CNN-DailyMail | 0.9875 | 0.8471 | ↓14.2% | 0.9943 | 0.9830 | ↓**1.1%** |
| IMDB | 0.9675 | 0.8675 | ↓10.3% | 0.9770 | 0.9451 | ↓**3.2%** |
| ROCStories | 0.9825 | 0.9412 | ↓4.2% | 0.9869 | 0.9865 | ↓**0.1%** |
| Average | 0.9712 | 0.8792 | ↓9.4% | 0.9793 | 0.9603 | ↓**1.9%** |

**Recovery-model Independence (API-based vs. Local LLMs).** We examine whether D&R depends on the choice of recovery model. In addition to DeepSeek-v3 (the API-based recovery model used in the main experiments), we evaluate Mistral-7B-Instruct-v0.3 as a locally deployed recovery model. As shown in Table 4, D&R maintains strong performance (mean AUROC 0.9614 vs. 0.9359), with only ∼2.5% degradation when switching from a large API model to a smaller local model. Importantly, D&R with Mistral-7B still outperforms RAIDAR even when RAIDAR relies on

the larger DeepSeek-v3 as the recovery model (data omitted for brevity). These results demonstrate that D&R is robust across recovery-model families and scales, and remains practically deployable even with smaller local models.

Table 4: AUROC performance with two Recovery Models: DeepSeek-v3 (API-based) and Mistral-7B-Instruct-v0.3 (local).

| Dataset | Source Model | DeepSeek-v3 *(API-based)* | | Mistral-7B *(Local)* |
|---|---|---|---|---|
| | | **D&R** | RAIDAR | **D&R** |
| ML-ArXiv-Papers | Qwen-Turbo | 0.8580 | 0.8375 | 0.8039 |
| | GPT4.1 | 0.9108 | 0.8600 | 0.8656 |
| | Gemini2.5-Flash | 0.9299 | 0.9025 | 0.8972 |
| | Grok3 | 0.9559 | 0.7700 | 0.8157 |
| CNN-DailyMail | Qwen-Turbo | 0.9800 | 0.8300 | 0.9800 |
| | GPT4.1 | 0.9908 | 0.9125 | 0.9844 |
| | Gemini2.5-Flash | 0.9901 | 0.9325 | 0.9862 |
| | Grok3 | 0.9856 | 0.8725 | 0.9784 |
| IMDB | Qwen-Turbo | 0.9584 | 0.8400 | 0.9381 |
| | GPT4.1 | 0.9456 | 0.8600 | 0.9289 |
| | Gemini2.5-Flash | 0.9890 | 0.9475 | 0.9713 |
| | Grok3 | 0.9688 | 0.9275 | 0.9398 |
| ROCStories | Qwen-Turbo | 0.9667 | 0.8725 | 0.9522 |
| | GPT4.1 | 0.9851 | 0.9150 | 0.9758 |
| | Gemini2.5-Flash | 0.9849 | 0.9500 | 0.9818 |
| | Grok3 | 0.9842 | 0.8875 | 0.9752 |
| Mean±SD | | **0.9614±0.0350** | 0.8823±0.0475 | **0.9359±0.0579** |

**Further Robustness and Generalization.** To thoroughly evaluate D&R's applicability, we extended our experiments to two additional settings: (i) **Adversarial Robustness**: On the RAID benchmark (Dugan et al., 2024), D&R retains high efficacy (AUROC 0.87) and proves resilient against 11 varying attack categories, most notably paraphrasing. (ii) **Multilingual Generalization**: Experiments on German, Spanish, and French confirmed that D&R generalizes effectively beyond English, achieving $> 0.93$ AUROC on long texts. Detailed results and analyses for these experiments are provided in **Appendix A.3**.

## 5 CONCLUSION

Disrupt-and-Recover (D&R) provides an efficient, black-box practical, and theoretically grounded framework for AI-text detection, achieving state-of-the-art accuracy and robustness across diverse settings, with particularly strong gains on short texts. Beyond these empirical results, D&R highlights posterior concentration as a guiding principle, opening new directions for disruption–recovery approaches across broader modalities and detection tasks. Detecting extremely short texts remains challenging as limited context obscures the concentration gap. Future work will address this via finer-grained disruption and retrieval-augmented signals to enhance sensitivity.

## ACKNOWLEDGEMENTS

We would like to thank the anonymous reviewers for their valuable feedback. This work was supported by the Hong Kong RGC General Research Fund (15221123 and 15216424), the PolyU Internal Fund (No. P0058468), the Huawei Gifted Fund, and the Key Laboratory of Smart Education of Guangdong Higher Education Institutes, Jinan University (2022LSYS003).

## ETHICS STATEMENT

All authors confirm adherence to the ICLR Code of Ethics. This work develops a detection method for AI-generated text to support research on transparency and reliability of language models. The method is designed as a technical tool to complement human judgment, and results should be interpreted with appropriate caution.

## REPRODUCIBILITY STATEMENT

We provide an anonymous repository with source code, datasets, and a requirements file specifying software dependencies. The experimental setup is described in the main text, and dataset sources and preprocessing are documented in Appendix A.6 and the repository. These resources are intended to facilitate independent verification of our findings.

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

## A  APPENDIX

### A.1  THE USE OF LARGE LANGUAGE MODELS (LLMS)

We acknowledge the use of large language models (ChatGPT and Gemini) as assistive tools in the preparation of this paper. Their role was strictly limited to language refinement, including grammar correction, sentence restructuring, and style polishing. All substantive research contributions—including hypothesis formulation, experimental design and execution, result analysis, and conclusions—are solely the work of the authors.

### A.2  DISCUSSION ON THE VALIDITY OF ASSUMPTIONS IN THEOREM 2

In this section, we provide a detailed breakdown of the assumptions underlying Theorem 2, discussing the specific conditions under which they hold and the rationale behind them.

**Assumption 1 (Upper Bound of Expected Similarity for Human-Written Text).**  There exist $\delta_0 > 0$ and $\epsilon > 0$ such that:

$$\mathbb{E}[S(\mathcal{T}_{orig}^{Human}, \mathcal{T}_{rec}^{Human})] < (1 - \delta_0)(1 - \omega(r_H)) \tag{1}$$

**When It Holds:** This assumption holds if the recovery distribution of human text satisfies two theoretical properties:

1. *Posterior concentration:* $\Pr(A) \geq 1 - \delta_H$, where $A = \{d(\mathcal{T}_{orig}^{Human}, \mathcal{T}_{rec}^{Human}) \leq r_H\}$, as per Theorem 1.

2. *Negligible excess similarity:* For the concentrated subset, the excess similarity is negligible, i.e., $\alpha = \mathbb{E}[S \mid A] - (1 - \omega(r_H)) = o(1 - \omega(r_H))$; and for the deviated subset, the similarity is negligible, i.e., $\beta = \mathbb{E}[S \mid A^c] = o(1 - \omega(r_H))$.

**Rationale:** Human text inherently lacks the specific pretraining biases of LLMs (often exhibiting more flexible semantics and diverse structures). Consequently, even well-recovered human text cannot achieve the "exact consistency" typical of AI text (justifying the bound on $\alpha$), while deviated recoveries typically result in near-zero similarity (justifying $\beta$). The expectation decomposition $\mathbb{E}[S_{Human}] = \mathbb{E}[S \mid A] \Pr(A) + \mathbb{E}[S \mid A^c] \Pr(A^c)$ mathematically derives the upper bound, ensuring theoretical rigor.

**Assumption 2 (Gap in Theoretical Lower Bounds for AI-Generated Text).**

$$(1 - \delta_A)(1 - \omega(r_A)) \geq (1 - \delta_H)(1 - \omega(r_H)) + 2\epsilon \tag{2}$$

**When It Holds:** This assumption holds for standard LLMs (e.g., GPT-4, Gemini) under normal generation settings (implying a high probability of occurrence):

1. *Practical condition:* AI text is generated with standard parameters (e.g., temperature $\leq 1.0$, without random token insertion).

2. *Theoretical condition:* $\delta_A \leq \delta_H$, $r_A \leq r_H$, and $2\epsilon$ is less than or equal to the intrinsic AI-human lower bound gap.

**Rationale:** LLMs are pretrained to optimize local token predictability. This objective leads to inherently stronger posterior concentration (smaller deviation radius $r_A$ and higher probability mass $1 - \delta_A$) compared to human text, creating a natural distributional gap.

**Assumption 3 (Compatibility Condition).**

$$\delta_H \geq \delta_0 \geq \delta_H - \frac{\epsilon}{1 - \omega(r_H)} \tag{3}$$

**When It Holds:** This assumption holds for **all valid parameter tunings** (providing a 100% chance of a non-empty interval):

1. *Practical condition:* $\epsilon$ is set to a small conceptual margin (e.g., 0.03–0.08), aligned with observed AI-human generation differences.

2. *Theoretical condition:* $\epsilon \leq \delta_H(1 - \omega(r_H))$. This is naturally satisfied since $\delta_H > 0$ for human text and $1 - \omega(r_H) > 0$ for any reasonable radius $r_H > 0$.

**Rationale:** This interval serves to balance Assumptions 1 and 2. A value for $\delta_0$ can always be chosen within this range (e.g., $\delta_0 = \delta_H - \frac{\epsilon}{2(1 - \omega(r_H))}$) to strictly avoid mathematical contradictions.

**Summary.** The assumptions may fail only in extreme edge cases (e.g., AI text generated by non-pretrained/random models, or human text intentionally mimicking AI patterns). However, they hold universally in standard AI-text detection tasks. Our sanity checks confirm their practical applicability, while the theoretical conditions ensure a high probability of occurrence in real-world scenarios.

## A.3 EXTENDED EXPERIMENTAL ANALYSIS

In this section, we present comprehensive evaluations concerning adversarial robustness, the ablation of disruption strategies, and multilingual generalization to further validate the effectiveness of D&R.

### A.3.1 ROBUSTNESS AGAINST ADVERSARIAL ATTACKS

To evaluate the robustness of D&R against adversarial attempts to evade detection, we utilized the **RAID** (Dugan et al., 2024). We tested D&R against 11 diverse attack types, ranging from character-level perturbations (e.g., homoglyphs) to high-level semantic obfuscations (e.g., paraphrasing).

As shown in Table 5, D&R maintains strong performance across all attack categories. Even under **Paraphrase** attacks—typically considered the most challenging for detection—D&R maintains a strong AUROC of 0.8210. Furthermore, for character-level attacks (e.g., Homoglyph, Zero Width Space), performance remains robust ($> 0.83$). These results indicate that our disruption-recovery mechanism relies on intrinsic posterior concentration rather than surface-level artifacts, making it difficult to fool via simple perturbations.

Table 5: Robustness of D&R on the RAID Dataset (AUROC). The method maintains high detection performance across various attack types.

| Attack Type | AUROC | Attack Type | AUROC |
|---|---|---|---|
| **None (Clean)** | **0.8736** | Homoglyph | 0.8352 |
| Insert Paragraphs | 0.8428 | Number | 0.8641 |
| Alternative Spelling | 0.8564 | Paraphrase | 0.8210 |
| Article Deletion | 0.8627 | Whitespace | 0.8505 |
| Synonym | 0.8139 | Upper/Lower | 0.8479 |
| Perplexity Misspelling | 0.8643 | Zero Width Space | 0.8322 |

### A.3.2 ABLATION STUDY OF WITHIN-CHUNK SHUFFLING AND RECOVERY SIMILARITY

To verify the necessity of our **Within-Chunk Shuffling (WCS)** strategy, we compared it against two alternative disruption mechanisms:

**Global Shuffling:** Randomly shuffling all tokens in the text.

**Chunk-Order Shuffling:** Shuffling the order of chunks while keeping tokens within chunks intact.

Table 6 presents the results across four datasets.

Table 6: Ablation Study Results comparison (Avg. AUROC on 4 Datasets).

| Disruption Method | ML-ArXiv | CNN-DM | IMDB | ROCStories | Avg. |
|---|---|---|---|---|---|
| D&R (Global Shuffling) | 0.5421 | 0.5833 | 0.5612 | 0.5390 | 0.5564 |
| D&R (Chunk-Order Shuffling) | 0.7130 | 0.7544 | 0.7205 | 0.7811 | 0.7423 |
| **D&R (WCS - Ours)** | **0.9266** | **0.9830** | **0.9451** | **0.9861** | **0.9602** |

- **Global Shuffling:** The severe disruption destroys all semantic context, making recovery impossible for both AI and Human texts. Since both fail to be recovered, they become indistinguishable, dropping performance to random guessing ($\sim$0.55).

- **Chunk-Order Shuffling:** Preserving internal token order makes the task trivial, allowing both AI and Human texts to be recovered with high fidelity. This "ceiling effect" causes their recoverability scores to converge, significantly reducing discriminability.

- **WCS (Ours):** WCS proves to be the optimal disruption strategy. It disrupts local token order to challenge the model while preserving semantic anchors, thereby maximizing the observable "concentration gap" between AI and Human text.

### A.3.3 MULTILINGUAL GENERALIZATION

To demonstrate that D&R is not limited to English, we extended our experiments to **German (DE), Spanish (ES), and French (FR)**. We utilized the MLSUM (Scialom et al., 2020) dataset for long texts and the Amazon (Keung et al., 2020) reviews dataset for short texts, averaging results across diverse source models.

As detailed in Table 8, D&R achieves consistently high performance across all tested languages (AUROC $> 0.93$ for long texts). Even on challenging short texts, it maintains robust performance ($> 0.83$). This confirms that the principle of posterior concentration is not an artifact of English-centric training but holds across different languages.

Table 7: Ablation study of D&R by removing Semantic or Structural Recovery Similarity.

| Dataset | Source Model | w/o SemanticSim | w/o StructuralSim | D&R |
|---|---|---|---|---|
| ML-ArXiv-Papers | Qwen-Turbo | 0.6529 | 0.7342 | 0.8580 |
| | GPT4.1 | 0.6616 | 0.7268 | 0.9108 |
| | Gemini2.5-Flash | 0.7029 | 0.7311 | 0.9299 |
| | Grok3 | 0.7272 | 0.7611 | 0.9559 |
| CNN-DailyMail | Qwen-Turbo | 0.6567 | 0.7465 | 0.9800 |
| | GPT4.1 | 0.6627 | 0.7151 | 0.9908 |
| | Gemini2.5-Flash | 0.6955 | 0.7731 | 0.9901 |
| | Grok3 | 0.6653 | 0.7072 | 0.9856 |
| IMDB | Qwen-Turbo | 0.7112 | 0.8560 | 0.9584 |
| | GPT4.1 | 0.7011 | 0.8551 | 0.9456 |
| | Gemini2.5-Flash | 0.7013 | 0.8806 | 0.9890 |
| | Grok3 | 0.7159 | 0.8716 | 0.9688 |
| ROCStories | Qwen-Turbo | 0.6581 | 0.7158 | 0.9667 |
| | GPT4.1 | 0.6561 | 0.7410 | 0.9851 |
| | Gemini2.5-Flash | 0.8022 | 0.7910 | 0.9849 |
| | Grok3 | 0.6796 | 0.7256 | 0.9842 |
| Average | | 0.6906 ↓**28.1%** | 0.7707 ↓**19.8%** | 0.9614 |

Table 8: Multilingual Performance (Avg. AUROC) on long (MLSUM) and short (Amazon) texts.

| Language | Long Text (MLSUM) | Short Text (Amazon) |
|---|---|---|
| German (DE) | 0.9306 | 0.8313 |
| Spanish (ES) | 0.9556 | 0.8604 |
| French (FR) | 0.9377 | 0.8592 |
| **Overall Avg** | **∼0.94** | **∼0.85** |

## A.4 EFFICIENCY COMPARISON

To quantify the practical benefits of our single-call framework, we compared the average latency and estimated cost of D&R against RAIDAR, the most competitive baseline which requires multiple generation calls.

As shown in Table 9, D&R drastically reduces computational overhead. Specifically, the single-call design lowers the average latency from 15 seconds to 2 seconds per sample and reduces the estimated API cost from $5 to $0.2 per 1,000 samples. This confirms that D&R is not only accurate but also highly efficient for large-scale deployment.

Table 9: Efficiency comparison between the multi-call baseline (RAIDAR) and our single-call method (D&R). Cost is estimated per 1,000 samples.

| Method | Avg Latency (s) | Est. Cost ($/1k samples) | Calls per Sample |
|---|---|---|---|
| RAIDAR | 15 | $5 | ∼5 calls |
| **D&R (Ours)** | **2** | **$0.2** | **1 call** |

## A.5 ADDITIONAL RESULTS

In this section, we present comprehensive performance data to supplement the main experimental results. Tables 10, 11, 12, and 13 provide the detailed AUROC breakdown on four long-text datasets, namely ML-ArXiv-Papers, CNN-DailyMail, IMDB, and ROCStories, respectively. This table expands upon the summarized results in the main text, demonstrating D&R's consistent superiority across diverse source models and text domains.Table 14 reports the TPR scores at fixed FPR threshold of **1%** and **5%**.

Table 10: AUROC on ML-ArXiv-Papers datasets across six source models.

| | | | | | | |
|---|---|---|---|---|---|---|
| **ML-ArXiv-Papers** | | | | | | |
| Source Model | RoBERTa | Likelihood | DNA-GPT | Fast-Detect | RAIDAR | Dald |
| GPT2 | 0.9333 | 0.9206 | 0.8679 | 0.9489 | 0.9033 | 0.9432 |
| GPT-Neo-2.7B | 0.6245 | 0.8193 | 0.8741 | 0.8334 | 0.8933 | **0.9786** |
| Qwen-Turbo | 0.5137 | 0.7112 | 0.6309 | 0.8038 | 0.8375 | 0.7210 |
| GPT4.1 | 0.5406 | 0.5352 | 0.4966 | 0.5878 | 0.8600 | 0.6326 |
| Gemini2.5-Flash | 0.5613 | 0.4703 | 0.4826 | 0.6115 | 0.9025 | 0.7477 |
| Grok3 | 0.5441 | 0.5206 | 0.4882 | 0.5596 | 0.7700 | 0.6798 |
| Source Model | OOD | ImBD | Binoculars | Text-Flu | DeTeCtive | **D&R(ours)** |
| GPT2 | 0.9608 | 0.6944 | 0.7856 | **0.9772** | 0.9433 | 0.9660 |
| GPT-Neo-2.7B | 0.8483 | 0.6103 | 0.7487 | 0.9701 | 0.8283 | 0.9390 |
| Qwen-Turbo | 0.7935 | 0.8192 | 0.7821 | 0.7980 | 0.7581 | **0.8580** |
| GPT4.1 | 0.6995 | 0.8259 | 0.5500 | 0.8286 | 0.7666 | **0.9108** |
| Gemini2.5-Flash | 0.6156 | 0.8220 | 0.5699 | 0.7817 | 0.6416 | **0.9299** |
| Grok3 | 0.6712 | 0.8439 | 0.4248 | 0.6038 | 0.7250 | **0.9559** |

Table 11: AUROC on four CNN-DailyMail datasets across six source models.

| | | | | | | |
|---|---|---|---|---|---|---|
| **CNN-DailyMail** | | | | | | |
| Source Model | RoBERTa | Likelihood | DNA-GPT | Fast-Detect | RAIDAR | Dald |
| GPT2 | 0.9392 | 0.8234 | 0.9126 | 0.8760 | 0.8367 | 0.8662 |
| GPT-Neo-2.7B | 0.6084 | 0.5390 | 0.7041 | 0.8760 | 0.8367 | 0.8486 |
| Qwen-Turbo | 0.5705 | 0.6163 | 0.5618 | 0.5675 | 0.8300 | 0.6876 |
| GPT4.1 | 0.5212 | 0.4796 | 0.4625 | 0.4240 | 0.9125 | 0.6441 |
| Gemini2.5-Flash | 0.5288 | 0.5339 | 0.4531 | 0.5419 | 0.9325 | 0.6741 |
| Grok3 | 0.5363 | 0.4799 | 0.4778 | 0.4249 | 0.8725 | 0.6808 |
| Source Model | OOD | ImBD | Binoculars | Text-Flu | DeTeCtive | **D&R(ours)** |
| GPT2 | 0.9000 | 0.7303 | 0.6822 | 0.9509 | 0.9000 | **0.9734** |
| GPT-Neo-2.7B | 0.7666 | 0.6284 | 0.5993 | 0.9495 | 0.7666 | **0.9734** |
| Qwen-Turbo | 0.6346 | 0.7485 | 0.5519 | 0.7617 | 0.6346 | **0.9800** |
| GPT4.1 | 0.8726 | 0.8852 | 0.4004 | 0.8876 | 0.8726 | **0.9908** |
| Gemini2.5-Flash | 0.9131 | 0.9592 | 0.5471 | 0.8606 | 0.9131 | **0.9901** |
| Grok3 | 0.8900 | 0.9175 | 0.4186 | 0.9324 | 0.8900 | **0.9856** |

Table 12: AUROC on four IMDB datasets across six source models.

| | | | | | | |
|---|---|---|---|---|---|---|
| **IMDB** | | | | | | |
| Source Model | RoBERTa | Likelihood | DNA-GPT | Fast-Detect | RAIDAR | Dald |
| GPT2 | 0.8579 | 0.8340 | 0.8314 | 0.9034 | 0.7833 | **0.9257** |
| GPT-Neo-2.7B | 0.7000 | 0.7867 | 0.8290 | 0.8115 | 0.8467 | 0.7451 |
| Qwen-Turbo | 0.5156 | 0.5901 | 0.5841 | 0.5862 | 0.8400 | 0.5509 |
| GPT4.1 | 0.5465 | 0.6174 | 0.5267 | 0.6257 | 0.8600 | 0.6072 |
| Gemini2.5-Flash | 0.5309 | 0.5398 | 0.5601 | 0.7422 | 0.9475 | 0.6834 |
| Grok3 | 0.5178 | 0.6027 | 0.5633 | 0.6975 | 0.9275 | 0.6525 |
| Source Model | OOD | ImBD | Binoculars | Text-Flu | DeTeCtive | **D&R(ours)** |
| GPT2 | 0.9646 | 0.7780 | 0.7611 | 0.7888 | 0.9183 | 0.9036 |
| GPT-Neo-2.7B | 0.8336 | 0.6828 | 0.7523 | 0.8357 | 0.8150 | **0.9056** |
| Qwen-Turbo | 0.8428 | 0.8739 | 0.5431 | 0.9424 | 0.8155 | **0.9584** |
| GPT4.1 | 0.8319 | 0.8493 | 0.5756 | 0.9040 | 0.8091 | **0.9456** |
| Gemini2.5-Flash | 0.7389 | 0.9442 | 0.7051 | 0.9403 | 0.8650 | **0.9890** |
| Grok3 | 0.8234 | 0.9261 | 0.6527 | 0.9388 | 0.8483 | **0.9688** |

Table 13: AUROC on ROCStories datasets across six source models.

| ROCStories | | | | | | |
|---|---|---|---|---|---|---|
| Source Model | RoBERTa | Likelihood | DNA-GPT | Fast-Detect | RAIDAR | Dald |
| GPT2 | 0.8563 | 0.8833 | 0.9199 | 0.8655 | 0.9856 | 0.8915 |
| GPT-Neo-2.7B | 0.8918 | 0.8533 | 0.8811 | 0.8862 | 0.9833 | 0.9613 |
| Qwen-Turbo | 0.6254 | 0.5385 | 0.5312 | 0.5625 | 0.8725 | 0.5025 |
| GPT4.1 | 0.6710 | 0.4999 | 0.5132 | 0.5105 | 0.9150 | 0.4411 |
| Gemini2.5-Flash | 0.7476 | 0.2809 | 0.4066 | 0.4888 | 0.9500 | 0.4229 |
| Grok3 | 0.7131 | 0.4553 | 0.4866 | 0.5175 | 0.8875 | 0.4311 |
| Source Model | OOD | ImBD | Binoculars | Text-Flu | DeTeCtive | **D&R***(ours)* |
| GPT2 | 0.9956 | 0.4360 | 0.4381 | 0.6024 | 0.9691 | **0.9970** |
| GPT-Neo-2.7B | 0.9982 | 0.6708 | 0.7982 | 0.6152 | 0.9883 | **0.9988** |
| Qwen-Turbo | 0.8448 | 0.7582 | 0.4974 | 0.8257 | 0.7533 | **0.9667** |
| GPT4.1 | 0.8689 | 0.7774 | 0.4225 | 0.8129 | 0.8316 | **0.9851** |
| Gemini2.5-Flash | 0.8833 | 0.8575 | 0.4501 | 0.7432 | 0.8916 | **0.9849** |
| Grok3 | 0.8745 | 0.8110 | 0.4261 | 0.8400 | 0.8199 | **0.9842** |

Table 14: **TPR (%) at Fixed FPR Thresholds.** Detailed performance breakdown across different source models for both long and short text settings.

| Dataset | Metric | Qwen | GPT | Gemini | Grok |
|---|---|---|---|---|---|
| Long Text | TPR@1%FPR | 74.8 | 81.8 | 90.8 | 79.5 |
| | TPR@5%FPR | 85.4 | 89.8 | 93.8 | 94.6 |
| Short Text | TPR@1%FPR | 49.8 | 57.6 | 53.8 | 47.9 |
| | TPR@5%FPR | 64.7 | 73.6 | 70.6 | 69.8 |

## A.6 DATASET DETAILS

**ML-ArXiv-Papers.** This dataset consists of abstracts from research papers in the computer science domain, particularly in machine learning, sourced from the ArXiv platform. The text is characterized by its professional language, rigorous structure, and strong logical coherence, representing a formal academic writing style.

**CNN-DailyMail.** Comprising news articles from CNN and Daily Mail, this dataset is rich in factual statements and coherent narrative structures. The text is of high quality and written in accessible language, making it a common benchmark for news summarization and text generation research.

**IMDB.** The IMDB dataset contains a large collection of user-written movie reviews. These texts are highly subjective, feature rich linguistic expression, and convey strong sentimental polarity and personalized styles.

**ROCStories.** This dataset is composed of five-sentence stories centered around everyday life scenarios. These texts exhibit clear narrative structures and causal relationships, embodying the characteristics of short, narrative-driven text.

**Wikihow.** This dataset contains texts extracted from "How-to" guides on wikiHow. The content is concise, presented in a formal style, and typically structured as clear, step-by-step instructions.

**AG-News.** Consisting of news headlines and short descriptions, this dataset exemplifies the style of short news text. It is highly condensed, formally structured, and logically coherent.

**Reddit.** This dataset is a collection of user-generated post titles and summaries from the Reddit platform. The language is colloquial and diverse in style, with a free and irregular structure that reflects the nature of social media communication.

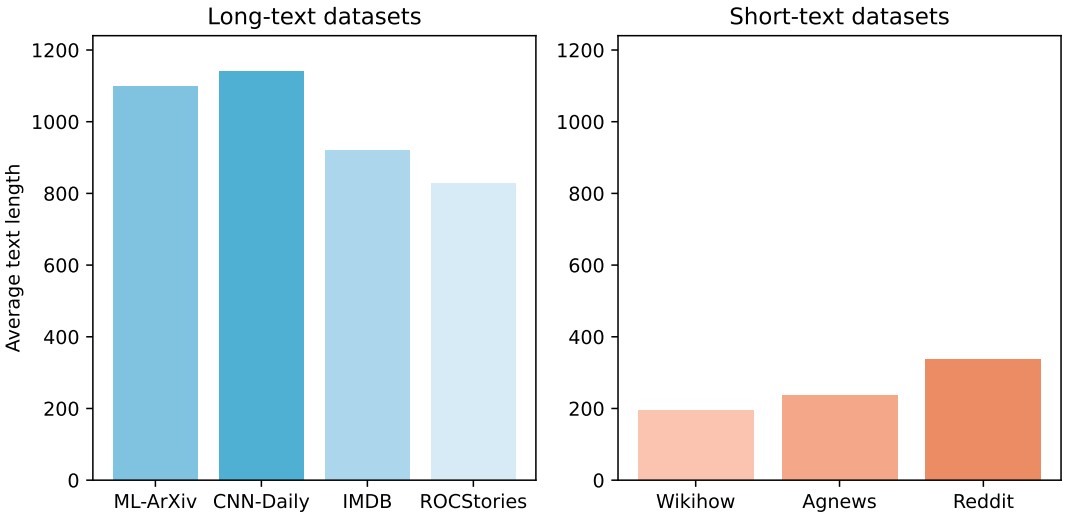

Figure 5: Average-text-length distributions for the long- and short-text datasets.

