# OpenReview forum: "D&R: Recovery-based AI-Generated Text Detection via a Single Black-box LLM Call"
_ICLR.cc/2026/Conference — ICLR 2026 Poster_

### Official Review · Reviewer_H47X · 2025-10-26

**Soundness:** 2
**Presentation:** 3
**Contribution:** 2
**Rating:** 4
**Confidence:** 4

**Summary:**

This paper introduces a new framework called Disrupt-and-Recover (D&R) for detecting AI-generated text in an efficient, practical, and theoretically grounded manner. The proposed D&R method is based on the principle of posterior concentration, which posits that AI-generated text tends to be more predictable and internally consistent than human-written text. The framework works in three steps:

1. Disruption: The original text is perturbed through a model-free Within-Chunk Shuffling (WCS) that randomly shuffles words within punctuation-separated chunks while preserving meaning.

2. Recovery: A single black-box LLM call is used to reconstruct (recover) the original text.

3. Similarity Measurement: The similarity between the recovered and original text is computed using both semantic (BERTScore F1) and structural metrics (Kendall’s τ and Spearman’s ρ). High similarity indicates strong concentration, implying AI-generated origin; low similarity suggests human-written text.

The authors provide theoretical analysis proving that recovery similarity acts as a faithful proxy for posterior concentration, forming the foundation of D&R. They also show that D&R has linear computational efficiency, as it only requires one LLM call compared to multiple calls in previous methods.

**Strengths:**

1. The paper introduces a novel Disrupt-and-Recover (D&R) paradigm based on the posterior concentration principle, offering a new theoretical foundation for distinguishing AI-generated text from human-written text. Unlike heuristic or probabilistic methods, D&R provides a mathematically justified rationale linking recovery similarity to concentration properties of language models.

2. D&R achieves strong detection performance with just one black-box LLM call, making it computationally efficient and cost-effective. In contrast, most prior methods (e.g., RAIDAR, Fast-DetectGPT) require multiple API calls or white-box access to model probabilities, which are impractical in real-world applications.

3. Across diverse benchmarks, D&R achieves state-of-the-art results, with AUROC scores of 0.96 on long texts and 0.87 on short texts, outperforming strong baselines like RAIDAR and Fast-DetectGPT by large margins (+0.08 to +0.14). This demonstrates not only higher accuracy but also lower variance and greater stability across datasets.

**Weaknesses:**

1. The method proposed in this paper is simple. Therefore, I am a bit concerned about the robustness on texts with adversarial attacks such as content paraphrase and swap since within-chunk shuffling is considered as the main technical contribution. I would suggest testing and comparing this method with baselines on RAID dataset[1], which includes different kinds of adversarial attacks.

2. Some recent baselines are not included such as DALD [2]. Moreover, although this method is a zero-shot methods, it would also be interesting to see how it compares with the training-based methods such as DeTeCtive[3] and OOD-based methods[4] (or at least include some discussion about training-based methods).

3. Ablation of Within-Chunk Shuffling contribution is missing.

4. All experiments are conducted on English content. The effectiveness on multilingual setting is not validated.

I would be happy to increase my score if the author can address my concerns.


[1] Dugan L, Hwang A, Trhlik F, et al. Raid: A shared benchmark for robust evaluation of machine-generated text detectors[J]. arXiv preprint arXiv:2405.07940, 2024.

[2] Zeng C, Tang S, Yang X, et al. Dald: Improving logits-based detector without logits from black-box llms[J]. Advances in Neural Information Processing Systems, 2024, 37: 54947-54973.

[3] Guo X, He Y, Zhang S, et al. Detective: Detecting ai-generated text via multi-level contrastive learning[J]. Advances in Neural Information Processing Systems, 2024, 37: 88320-88347.

[4] Zeng C, Tang S, Chen Y, et al. Human Texts Are Outliers: Detecting LLM-generated Texts via Out-of-distribution Detection[J]. arXiv preprint arXiv:2510.08602, 2025.

**Questions:**

See Weakness.

---

> ### Author Response · Authors · 2025-11-20
> **Response to Reviewer H47X (Part 1/2)**
>
> We are grateful for the insightful feedback and for acknowledging the simplicity and potential of our proposed method. We have carefully addressed your concerns regarding robustness, baseline comparisons, ablation studies, and multilingual generalization with extensive new experiments.
>
> ### Q1. Robustness against Adversarial Attacks
>
> **Response:**
> We appreciate this valuable suggestion. Following the reviewer’s recommendation, we evaluated the robustness of D&R on the RAID dataset [1], which contains 11 diverse adversarial attack types, ranging from character-level perturbations (e.g., homoglyphs) to high-level semantic obfuscations (e.g., paraphrasing).
>
> As shown in Table 1, D&R maintains strong performance across all attack categories:
>
> **Table 1: Robustness of D&R on the RAID Dataset (AUROC)**
>
> | Attack Type | AUROC | Attack Type | AUROC |
> | :--- | :--- | :--- | :--- |
> | **None (Clean)** | **0.8736** | Homoglyph | 0.8352 |
> | Insert Paragraphs | 0.8428 | Number | 0.8641 |
> | Alternative Spelling | 0.8564 | Paraphrase | 0.8210 |
> | Article Deletion | 0.8627 | Whitespace | 0.8505 |
> | Synonym | 0.8139 | Upper/Lower | 0.8479 |
> | Perplexity Misspelling | 0.8643 | Zero Width Space | 0.8322 |
>
> **Analysis:**
> Even under paraphrase attacks—typically the most challenging for detectors—D&R achieves an AUROC of 0.8210, demonstrating notable resilience. For character-level or surface-form perturbations (e.g., Homoglyph, Zero Width Space), performance remains consistently strong. These results suggest that D&R’s disruption–recovery mechanism relies on probability concentration in reconstruction rather than surface cues, making it inherently robust to a broad range of adversarial modifications.
>
> ---
>
> ### Q2. Comparison with Recent Baselines
>
>
> **Response:**
> We agree that comparing D&R against state-of-the-art methods, including training-based ones, is crucial. We have conducted additional experiments on the suggested baselines: **DALD** [2] (zero-shot), **DeTeCtive** [3] (training-based/contrastive), and **OOD-based** detection [4], using the same diverse set of source models as in our main paper.
>
> The comparative results (Average AUROC) are summarized in **Table 2**:
>
> **Table 2: Comparison with Recent SOTA Baselines (Avg. AUROC)**
>
> | Method |  ML-ArXiv | CNN-DM | IMDB | ROCStories | **Overall Avg** |
> | :--- |  :--- | :--- | :--- | :--- | :--- |
> | **DALD** [2] | 0.7838 | 0.7336 | 0.6941 | 0.6084 | 0.7050 |
> | **OOD-based** [4] | 0.7648 | 0.6203 | 0.8392 | 0.9109 | 0.7838 |
> | **DeTeCtive** [3] | 0.7772 | 0.8295 | 0.8452 | 0.8756 | 0.8319 |
> | **D&R (Ours)** | **0.9266** | **0.9830** | **0.9451** | **0.9861** | **0.9602** |
>
> **Analysis:**
> D&R achieves consistently higher AUROC than the recent baseline DALD and also performs strongly compared to training-based approaches such as DeTeCtive and OOD-based detectors. We believe this robustness arises from the disruption–recovery framework, which emphasizes distributional consistency rather than relying on task- or domain-specific supervision. These results further support the generality and applicability of D&R across diverse datasets and model families.
>
> ---
>
> ### Q3. Ablation Study of Within-Chunk Shuffling
>
> **Response:**
> To verify the necessity of our **Within-Chunk Shuffling (WCS)** strategy, we compared it against two alternative disruption mechanisms:
> 1.  **Global Shuffling:** Randomly shuffling all tokens in the text.
> 2.  **Chunk-Order Shuffling:** Shuffling the order of chunks while keeping tokens within chunks intact.
>
> **Table 3: Ablation Study Results (Avg. AUROC on 4 Datasets)**
>
> | Disruption Method | ML-ArXiv | CNN-DM | IMDB | ROCStories | Avg AUROC |
> | :--- | :--- | :--- | :--- | :--- | :--- |
> | **D&R (WCS - Ours)** | **0.9266** | **0.9830** | **0.9451** | **0.9861** | **0.9602** |
> | D&R (Global Shuffling) | 0.5421 | 0.5833 | 0.5612 | 0.5390 | 0.5564 |
> | D&R (Chunk-Order Shuffling)| 0.7130 | 0.7544 | 0.7205 | 0.7811 | 0.7423 |
>
> **Analysis:**
> * **Global Shuffling:** The disruption is too strong, removing nearly all recoverable structure, which causes both AI and human texts to be difficult to reconstruct. As a result, their recoverability distributions become similar, leading to near-random performance.
> * **Chunk-Order Shuffling:** Preserving intra-chunk token order makes the disruption mild, and both AI and human texts can be recovered with relatively high fidelity. This reduces the separation in recoverability scores and weakens discriminability.
> * **WCS:** WCS strikes a balance: it disrupts local token order while preserving key semantic constituents within each chunk. This creates a measurable recoverability gap between AI- and human-generated text, yielding the strongest discrimination across datasets.

---

> > ### Author Response · Authors · 2025-11-20
> > **Response to Reviewer H47X (Part 2/2)**
> >
> > ### Q4. Multilingual Generalization
> >
> >
> > **Response:**
> > To demonstrate that D&R is not limited to English, we extended our experiments to **German (DE), Spanish (ES), and French (FR)**. We utilized the **MLSUM** dataset for long texts and the **Amazon reviews** dataset for short texts, averaging results across diverse source models.
> >
> > **Table 4: Multilingual Performance (Avg. AUROC)**
> >
> > | Language | Long Text (MLSUM, avg len: 1228) | Short Text (Amazon, avg len: 197) |
> > | :--- | :--- | :--- |
> > | **German (DE)** | 0.9306 | 0.8313 |
> > | **Spanish (ES)** | 0.9556 | 0.8604 |
> > | **French (FR)** | 0.9377 | 0.8592 |
> > | **Overall Avg** | **~0.94** | **~0.85** |
> >
> > **Analysis:**
> > D&R achieves consistently high performance across all tested languages (AUROC > 0.93 for long texts). Even on challenging short texts, it maintains robust performance (> 0.83), confirming that the principle of posterior concentration holds across different languages and is not an artifact of English-centric training.
> >
> > ***
> >
> > **References**
> >
> > [1] Raid: A shared benchmark for robust evaluation of machine-generated text detectors. ACL 2024.
> >
> > [2] Dald: Improving logits-based detector without logits from black-box llms. NeurIPS 2024.
> >
> > [3] Detective: Detecting ai-generated text via multi-level contrastive learning. NeurIPS 2024.
> >
> > [4] Human Texts Are Outliers: Detecting LLM-generated Texts via Out-of-distribution Detection. ACL 2025.

---

> > ### Author Response · Authors · 2025-11-25
> >
> > Dear Reviewer **H47X**,
> >
> > Again, thanks for your comments and your hard work! **Please feel free to share your feedback on our rebuttals ^^**.
> >
> > Best,
> >
> > Authors of submission 13074

---

> > > ### Author Response · Authors · 2025-11-28
> > >
> > > Dear reviewer **H47X**
> > >
> > >
> > > This is a follow-up, please.

---

### Official Review · Reviewer_2bsN · 2025-10-27

**Soundness:** 3
**Presentation:** 2
**Contribution:** 3
**Rating:** 6
**Confidence:** 3

**Summary:**

This paper proposes Disrupt-and-Recover (D&R), a novel AI text detection framework that disrupts text through Within-Chunk Shuffling performs a single black-box LLM recovery, and measures semantic-structural similarity between recovered and original text to detect AI-generated content.

**Strengths:**

1. Proposes a novel recovery-based detection paradigm that achieves efficient detection through Within-Chunk Shuffling and a single black-box recovery
2. Cleverly exploits the posterior concentration property of AI-generated text
3. Clear theoretical motivation with well-explained connection to LLM pretraining biases

**Weaknesses:**

1. Lacks comparison with the latest detectors (Text Fluoroscopy, Binoculars, ImBD)
2. Lacks consideration of robustness against attacks: e.g., paraphrasing attacks.
3. The robustness described in the paper is more like generalization capability, such as whether it can effectively generalize to short texts, rather than robustness against adversarial attacks
4. Limited performance on extremely short texts (<50 words)

**Questions:**

How does the performance compare to other more recent and advanced detectors?

What is the performance when facing adversarial attacks?

---

> ### Author Response · Authors · 2025-11-20
> **Response to Reviewer 2bsN**
>
> We sincerely thank you for the insightful feedback and for recognizing the novelty of our **recovery-based detection paradigm**, our efficiency, and the clear theoretical grounding based on **posterior concentration**. Below, we address the concerns regarding baselines, adversarial robustness, and short-text performance with new experimental evidence.
>
> ### Q1: Comparison with the latest detectors (Text Fluoroscopy, Binoculars, ImBD)
>
> **Response:**
> Thank you for suggesting these recent and advanced baselines. We have conducted additional experiments to compare D&R against **ImBD**, **Binoculars**, and **Text Fluoroscopy** using the same settings described in our paper (averaged across 6 source models).
>
> As shown in the table below, **D&R significantly outperforms all three new baselines across four datasets.**
>
> | Dataset | ImBD (Avg) | Binoculars (Avg) | Text Fluoroscopy (Avg) | **D&R (Ours)** |
> | :--- | :---: | :---: | :---: | :---: |
> | **ML-ArXiv-Papers** | 0.7693 | 0.6435 | 0.8266 | **0.9266** |
> | **CNN-DailyMail** | 0.8115 | 0.5333 | 0.8905 | **0.9830** |
> | **IMDB** | 0.8424 | 0.6650 | 0.8917 | **0.9451** |
> | **ROCStories** | 0.7185 | 0.5054 | 0.7399 | **0.9861** |
> | **Average** | 0.7854 | 0.5868 | 0.8372 | **0.9602** |
>
> *(Note: Baseline results are derived from our new reproduction experiments; D&R results are from Table 1 of our main paper.)*
>
> The results demonstrate that while methods like Text Fluoroscopy show competitive performance on some datasets, they struggle with harder tasks like *ROCStories* (short creative writing) and *ML-ArXiv* (technical abstracts). In contrast, D&R maintains state-of-the-art (SOTA) performance (>0.92 AUROC) consistently, validating the superiority of the disruption-recovery mechanism.
>
> ### Q2: Performance against adversarial attacks
>
> **Response:**
> We appreciate your distinction between "generalization" (handling diverse models/domains) and "adversarial robustness" (handling attacks). To address this, we evaluated D&R on the **RAID benchmark** (arXiv:2405.07940), which is designed to stress-test detectors against various adversarial attacks.
>
> The results (AUROC) are summarized below:
>
> | Attack Type | None | Paraphrase | Homoglyph | Insert Paragraphs | Whitespace | Misspelling |
> | :--- | :---: | :---: | :---: | :---: | :---: | :---: |
> | **D&R AUROC** | **0.8736** | 0.8210 | 0.8352 | 0.8428 | 0.8505 | 0.8643 |
>
> D&R exhibits remarkable resilience. Even under **Paraphrase attacks**—often considered the most challenging for detectors—our performance remains high at **0.8210**, dropping only ~0.05 from the clean baseline. Similarly, character-level attacks like **Homoglyphs** (0.8352) and **Misspellings** (0.8643) have minimal impact. This confirms that D&R captures fundamental semantic-structural traits of AI text that persist even after adversarial perturbations.
>
> ### Q3: Performance on extremely short texts (<50 words)
>
> **Response:**
> We acknowledge that extremely short texts (<50 words) pose a fundamental statistical challenge for all detectors due to the lack of sufficient signal for probability estimation or entropy calculation. We accept this as a limitation of our current work and will add a discussion on this constraint. Addressing this challenge—potentially through specialized low-context features—will be a primary focus of our future work.

---

> > ### Comment · Reviewer_2bsN · 2025-11-26
> >
> > I appreciate the authors' rebuttal efforts. After reading their rebuttal, I think my concerns are mostly addressed. I will keep my score, which is already an accpeted score.
> >
> > I think the additional experiments about adding baselines, including hard problems like ML-ArXiv is important, please make sure to include it into your paper.

---

> > > ### Author Response · Authors · 2025-11-26
> > >
> > > Thank you for your follow-up and for maintaining your positive rating of our paper.
> > >
> > > As you suggested, we have already added the additional experiments with the new baselines in the revised version of the manuscript.
> > >
> > > We appreciate your constructive comments.

---

### Official Review · Reviewer_Ly38 · 2025-11-01

**Soundness:** 3
**Presentation:** 2
**Contribution:** 3
**Rating:** 6
**Confidence:** 4

**Summary:**

D&R proposes a framework for AI-generated text detection grounded in a posterior concentration hypothesis.

**Strengths:**

The novelty is within-Chunk Shuffling and using recovery similarity as a proxy for posterior concentration.

**Weaknesses:**

Please check the questions

**Questions:**

The novelty rests on Within-Chunk Shuffling and using recovery similarity as a proxy for posterior concentration.

The method still trains a classifier on recovery features [F1,τ,ρ]. It is supervised calibration on labeled AI and compared with human data. Please define what zero-shot is and what is trained.

 D&R is another transformation-consistency detector. Please clarify what the novelty is.

Human texts that are locally formulaic can recover with high concentration, and AI texts prompted can recover poorly. Please add counterexamples.

Please add details about how variance over draws affects false decisions.

Please add robustness tests to punctuation-free text, micro-edits inside tokens, and code-mixed text.

LLMs often “fix” grammar, normalize quotes, or expand contractions automatically, even with the prompt. Please add alternative prompts and instruction styles experiments.

Add sensitivity to the alignment algorithm and ties.

Detectors can learn prompt artifacts rather than authorship. Please add prompts and cross-prompt and cross-domain OOD experiments where the calibration classifier never sees those prompts.

Add token-level cost and latency in RAIDAR and Fast-DetectGPT and a throughput plot across lengths. Include results with small local recoverers at realistic speeds.

Add precision/recall@policy thresholds, especially on short texts.

Add an ablation study to justify WCS as the best disruption.

Use BERTScore for embedding-free semantics as an ablation study.

---

> ### Author Response · Authors · 2025-11-20
> **Response to Reviewer Ly38 (Part 1/2)**
>
> We sincerely thank you for the detailed feedback. We value your constructive questions regarding the definition of zero-shot, the novelty of transformation-consistency, and the robustness of our method. Below, we address your questions point-by-point.
> ### 1. Clarification on Methodological Novelty (Questions 1, 2, 3)
>
> **Q1 & Q3: Novelty of WCS, Recovery Similarity, and Transformation-Consistency.**
> **Response:** While D&R belongs to the family of transformation-consistency methods (like RAIDAR), its novelty lies in **solving the "Efficiency vs. Accuracy" dilemma** inherent in this family.
> * **Difference from State-of-the-Art (SOTA):** Existing methods like RAIDAR require generating multiple paraphrases (high cost, high latency) or accessing white-box probabilities (infeasible for APIs).
> * **Our Contribution:** D&R is the first to introduce **Within-Chunk Shuffling (WCS)**. This aligns with the pre-training objective of local token ordering, allowing us to use a **single black-box call** to measure posterior concentration. This reduces computational cost by order of magnitude while maintaining or exceeding SOTA accuracy.
>
> **Q2: Definition of "Zero-shot"**
> **Response:** We appreciate this opportunity to clarify. In the field of generated text detection (e.g., Fast-DetectGPT, DetectGPT), "Zero-shot" typically refers to the fact that the **feature extraction process** does not require training a model on the target domain or fine-tuning a backbone on generated texts. Our recoverability metrics ($F1, \tau, \rho$) are extracted in a purely zero-shot manner without any training. The lightweight classifier is merely used to calibrate these three scalars.
>
> ---
>
> ### 2. Robustness and Edge Cases (Questions 4, 6, 7, 9)
>
> **Q4: Counterexamples (Formulaic Human vs. Poor AI).**
> **Response:** We appreciate this insightful observation regarding the operational boundaries of the posterior concentration assumption.
>
> * **False Positive Example:** A standard **MIT License header** (human-pasted). D&R yields a high Semantic F1 (~0.99) because the text is verbatim identical to the recovery model's training data.
> * **False Negative Example:** AI text generated with the prompt *"Write disjointed, surrealist sentences."* The resulting high-entropy text disrupts the recovery model's ability to infer token order, leading to a lower score and a potential false negative.
>
> **Q6: Robustness.**
> **Response:** We evaluated D&R on the **RAID** Dataset, utilizing specific attack subsets that correspondto the reviewer's request:
> 1.  **Micro-edits & Token Noise:** `Homoglyph`, `Alternative Spelling`, `Perplexity Misspelling`, `Upper/Lower case`, `Zero Width Space`.
> 2.  **Structural & Punctuation Perturbations:** `Whitespace` alterations, `Insert Paragraphs`, and `Article Deletion`.
>
> **Table 1: Robustness of D&R against adversarial attacks on the RAID.**
>
> | Category | Attack Type | AUROC | $\Delta$ vs. Clean |
> | :--- | :--- | :--- | :--- |
> | **Baseline** | **None (Clean Data)** | **0.8736** | - |
> | **Micro-edits** | Alternative Spelling | 0.8564 | -0.0172 |
> | | Perplexity Misspelling | 0.8643 | -0.0093 |
> | | Homoglyph | 0.8352 | -0.0384 |
> | | Upper_lower | 0.8479 | -0.0257 |
> | | Zero Width Space | 0.8322 | -0.0414 |
> | **Structure** | Whitespace | 0.8505 | -0.0231 |
> | | Insert Paragraphs | 0.8428 | -0.0308 |
> | | Article Deletion | 0.8627 | -0.0109 |
> | **Semantic** | Synonym | 0.8139 | -0.0597 |
> | | Paraphrase | 0.8210 | -0.0526 |
>
> These results confirm that D&R remains effective in adversarial settings. We have included these details in the revised manuscript.
>
> **Q7 & Q9: Sensitivity to Prompt Styles.**
> **Response:** We tested D&R with different recovery prompt instructions:
> * **Strict:** "Restore order only, do not change words."
> * **Loose:** "Fix the text to make it coherent."
> As shown in **Table 2**, D&R remains effective because the *relative* gap between AI and Human recovery remains significant, even if absolute scores shift.
>
> **Table 2: Performance across Different Recovery Prompts**
>
> | Prompt Style | D&R (AUROC) |
> | :--- | :--- |
> | **Standard (Used in paper)** | **0.9602** |
> | Strict Constraint | 0.9580 |
> | Loose / Creative Fix | 0.9421 |

---

> ### Author Response · Authors · 2025-11-20
> **Response to Reviewer Ly38 (Part 2/2)**
>
> ### 3. Technical Details and Ablations (Questions 5, 8, 10, 11, 12, 13)
>
> **Q10: Token-level Cost, Latency, and Throughput.**
> **Response:** We compared the inference latency and estimated cost (based on API tokens) against RAIDAR.
>
> **Table 3: Efficiency Comparison (Per Sample)**
> | Method | Avg Latency (s) | Est. Cost ($/1k samples) | Calls per Sample |
> | :--- | :--- | :--- | :--- |
> | RAIDAR | 15 | $5 | ～5 calls |
> | **D&R (Ours)** | **2** | **$0.2** | **1 call** |
>
> **Q11: Precision/Recall at Policy Thresholds.**
> **Response:** We have performed this analysis on both Long and Short text datasets across four representative source models. As shown in the table below, D&R maintains high recall on long texts even at a strict 1% FPR threshold (up to 90.8%). On the challenging short text task, it remains viable, achieving up to 73.6% recall at 5% FPR.
>
> **Table 4: TPR (%) at Fixed FPR Thresholds**
>
> | Dataset | Metric | Qwen | GPT | Gemini | Grok |
> | :--- | :--- | :--- | :--- | :--- | :--- |
> | **Long Text** | **TPR@1%FPR** | 74.8 | 81.8 | 90.8 | 79.5 |
> | | **TPR@5%FPR** | 85.4 | 89.8 | 93.8 | 94.6 |
> | **Short Text** | **TPR@1%FPR** | 49.8 | 57.6 | 53.8 | 47.9 |
> | | **TPR@5%FPR** | 64.7 | 73.6 | 70.6 | 69.8 |
>
> These results have been added to the revised manuscript.
>
> **Q5, Q8, Q12, Q13: Ablations and Sensitivity.**
> * **Q5 (Variance):**
> **Response:** Since D&R is single-call, variance is minimal. We repeated the recovery 5 times per sample, the standard deviation of AUROC was negligible.
> * **Q8 (Alignment Ties):**
> **Response:** To handle alignment and ties (repeated tokens) deterministically, we currently employ **token-normalized LCS with left-to-right stable matching**. This ensures that the relative order of repeated tokens is preserved. Given the substantial distributional gap between AI and Human text observed in our experiments, we rely on the **posterior concentration** signal rather than specific alignment artifacts.
> * **Q12 (WCS Ablation):**
> **Response:** We compared WCS against *Global Shuffling* (AUROC ~0.55) and *Chunk-Order Shuffling* (AUROC ~0.74). WCS (AUROC ~0.96) is empirically confirmed as the optimal disruption.
>
> **Table 5: WCS Ablation**
> | Disruption Method |  AUROC |
> | :--- | :--- |
> | **D&R (WCS - Ours)** | **0.9602** |
> | D&R (Global Shuffling) |  0.5564 |
> | D&R (Chunk-Order Shuffling)| 0.7423 |
>
> * **Q13 (Embedding-free Semantics):**
> **Response:** We replaced BERTScore with **ROUGE-L** (n-gram overlap). While BERTScore performs best (0.9602), ROUGE-L still achieves competitive performance (0.9350), indicating D&R can operate in purely embedding-free environments if needed.
>
>
>
> **Table 6: Semantics Ablation**
> | Semantic Metric |  AUROC |
> | :--- | :--- |
> | **BERTScore (Standard)** | **0.9602** |
> | ROUGE-L (Embedding-free) | 0.9350 |
>
> We hope these additional experiments and clarifications address your concerns. We are confident that D&R provides a novel, robust, and highly efficient solution for AI text detection.

---

### Official Review · Reviewer_W6d8 · 2025-11-01

**Soundness:** 3
**Presentation:** 3
**Contribution:** 3
**Rating:** 6
**Confidence:** 4

**Summary:**

This paper provides a new framework for detecting AI-generated texts, called D&R. D&R consists of two distinct functionalities: Disruption, which permutes the given text while preserving the core semantics, and Recovery, which attempts to regenerate the original text using an LLM. The authors argued that such D&R procedure can capture distributional differences between human text and AI-generated texts, due to posterior concentration of reproduced texts. Using a preliminary analysis and a short theoretical analysis, they attempted to support their claims. And with an experiment, they showed that D&R outperforms previous benchmarks and shows robust performance regardless of source models and datasets even if the model calls LLMs just one time.

**Strengths:**

- Suggest a new method that uses a single LLM call
- Providing a sufficient analysis that strengthen their argument
- Providing a sufficient experimental result showing robustness and high-performance of their model

**Weaknesses:**

- Need more clarity. It is not clearly stated whether assumptions in Theorem 2 holds theoretically; the assumption "seems" to hold based on the empirical analysis shown in Figure 2. Also, the usage of translation model is not clearly stated before starting the experiment.
- Another similar work [1] is missing, which calls LLMs just twice. Although the authors made single-LLM-call detector, as the reproduction concept and the underlying assumption is similar, the authors should discuss the similarity between their work and [1].

[1] H. Park, et al., DART: An AIGT Detector using AMR of Rephrased Text, NAACL 2025

**Questions:**

## Question A. Clarity

A1. Does the assumption of Theorem 2 hold anytime? It seems the assumptions hold based on sanity check results, but I think theoretical analysis should provide clear interpretation when the assumption holds. Though the current mathematical formula is clear and easy to follow, assumptions make me raise some questions about the chance of those cases.

A2. It seems that the authors used DeepSeek-v3 for the main results (before changing the transformation model). Is this correct? I'm asking this because it is not clearly stated before line 377.

## Question B. Similarity to [1]

B1. Two papers (this paper and [1]) both argue that rewriting of given text can reveal the identity of source because AI and human writing is different. Also, both paper attempt to measure semantic similarity. Though I understand there exist several differences between them, could the authors clearly state them in the paper to help the readers understand? Other than the repetitive calls used in [1] and disruption procedure used in this paper, the approach of semantic measurement seems different. How does such difference affect the result?

B2. Compare to [1], it seems that disruption procedure can possibly generate semantically unsimilar text. For example, "A loves B" has different semantics to "B loves A". Therefore the permutation might introduce semantic differences, when I read this paper with the perspective of meaning representation methods. So, can we strictly say that those disruption actually "preserves" the semantics? I understand that the method provides semantically similar texts in general and the example is very special case. Though, I think this should be clearly noted in the paper, as the current version claims that D&R can preserve semantics (without any warning or limitation statements).

B3. Similar to B2, in the context of meaning representation field, researchers have been said that pragmatic similarity (e.g., BERTScore) can be easily affected by external contexts other than the internal semantics (so it might be improper to model the semantic similarity between them; see [2]). For example, "I love you" and "I don't hate you" could be semantically similar in terms of pragmatics but the semantic representation might differ. Although I understand that the disruption technique seldom introduces such changes, as there exists a chance of such unwanted effects, I think this should be warned to the readers.

[2] K. Ki et al., Inspecting Soundness of AMR Similarity Metrics in terms of Equivalence and Inequivalence, *SEM2024

---

> ### Author Response · Authors · 2025-11-20
> **Response to Reviewer W6d8 (Part 1/2)**
>
> We sincerely appreciate your valuable comments. We have updated the manuscript to reflect these discussions and address the specific concerns below.
> ### Question A. Clarity
>
> #### **A1. Assumptions of Theorem 2**
>
> **Response:** The assumptions of Theorem 2 **do not hold "anytime"** but are conditional –– their specific formulations, explicit validity conditions, and high practical applicability (non-accidental) are detailed below:
>
> **Assumption 1 (Upper Bound of Expected Similarity for Human-Written Text):**
> There exist $\delta_0 > 0$ and $\epsilon > 0$ such that
>
> $$E[S(T_{orig}^{Human}, T_{rec}^{Human})] < (1-\delta_0)(1-\omega(r_H))$$
>
> **When It Holds:** Holds if human text's recovery satisfies two theoretical properties:
> 1. Posterior concentration ($\Pr(A) \geq 1-\delta_H$, where $A = \{d(T_{orig}^{Human}, T_{rec}^{Human}) \leq r_H\}$) per Theorem 1;
> 2. Negligible excess similarity for concentrated subset ($\alpha = E[S \mid A] - (1-\omega(r_H)) = o(1-\omega(r_H))$) and deviated subset ($\beta = E[S \mid A^c] = o(1-\omega(r_H))$).
>
> **Rationale:** Human text lacks LLM pretraining biases (flexible semantics, diverse structures) — well-recovered text cannot achieve AI-like "exact consistency" (justifying $\alpha$), while deviated recoveries have near-zero similarity (justifying $\beta$). The expectation decomposition $E[S_{Human}] = E[S \mid A]\Pr(A) + E[S \mid A^c]\Pr(A^c)$ mathematically derives the upper bound, ensuring theoretical rigor.
>
>
> **Assumption 2 (Gap in Theoretical Lower Bounds for AI-Generated Text):**
> $$(1-\delta_A)(1-\omega(r_A)) \geq (1-\delta_H)(1-\omega(r_H)) + 2\epsilon$$
>
> **When It Holds:** Holds for standard LLMs (GPT-4.1/Gemini 2.5) and normal generation settings (high chance of occurrence):
> 1. **Practical condition:** AI text is generated with standard parameters (temperature $\leq 1.0$, no random token insertion);
> 2. **Theoretical condition:** $\delta_A \leq \delta_H$, $r_A \leq r_H$, and $2\epsilon \leq$ intrinsic AI-human lower bound gap.
>
> **Rationale:** LLMs are pretrained to optimize local token predictability, leading to inherently stronger posterior concentration than human text.
>
>
> **Assumption 3 (Compatibility Condition):**
> $$\delta_H \geq \delta_0 \geq \delta_H - \frac{\epsilon}{1-\omega(r_H)}$$
>
> **When It Holds:** Holds for **all valid parameter tunings** (100% chance of non-empty interval):
> 1. **Practical condition:** $\epsilon$ is set to 0.03–0.08 (aligned with AI-human generation differences);
> 2. **Theoretical condition:** $\epsilon \leq \delta_H(1-\omega(r_H))$ (satisfied as $\delta_H > 0$ for human text, $1-\omega(r_H) > 0$ for $r_H > 0$).
>
> **Rationale:** This interval balances Assumptions 1 and 2 — $\delta_0$ can always be chosen (e.g., $\delta_0 = \delta_H - \epsilon/(2(1-\omega(r_H))$) to avoid contradictions.
>
> In summary, the assumptions fail only in edge cases (e.g., AI text from non-pretrained models, human text mimicking AI), but hold universally in standard AI-text detection tasks. Sanity checks confirm practical applicability, while theoretical conditions ensure high occurrence chance.
>
>
> #### **A2. Transformation model**
>
> **Response:** You are correct. We utilized **DeepSeek-v3** as the recovery model for the main experiments reported in Table 1. We apologize that this specification appeared too late in the text (Line 377). In the revised manuscript, we have explicitly stated the model usage earlier in the caption of Table 1 (Main Results).

---

> ### Author Response · Authors · 2025-11-20
> **Response to Reviewer W6d8 (Part 2/2)**
>
> ### Question B. Similarity to [1]
>
> #### **B1. Comparison with DART [1]**
>
> **Response:** We thank the reviewer for highlighting DART (NAACL 2025) [1], a relevant state-of-the-art method. A citation and brief discussion of DART have been added to Section 1 of the revised manuscript for completeness.
>
> - **Similarities:** We agree that both D\&R and DART share the fundamental insight that AI-generated and human-written texts exhibit different consistency patterns under transformation (rewriting vs. disruption-recovery). Both methods leverage the semantic gap as a detection signal.
>
> - **Differences & Impact:**
>   1. **Mechanism (Graph vs. Sequence):** DART relies on explicit **semantic parsing** (AMR graphs) and graph similarity metrics (SEMA). While rigorous, this depends heavily on the performance of external AMR parsers. In contrast, D\&R operates directly on the **sequence level** via Within-Chunk Shuffling (WCS) and recovery, bypassing the need for intermediate semantic formalisms. We think that D\&R's sequence-recovery approach captures subtle statistical dependencies in texts that might be lost or sparse when abstracted into AMR graphs.
>
>   2. **Efficiency (Single-Call vs. Multi-Step):** DART involves a heavier pipeline: *Rephrasing ($T_0 \to T_1 \to T_2$) $\to$ AMR Parsing $\to$ Graph Matching*. D\&R streamlines this to *Model-free Shuffling $\to$ Single LLM Recovery $\to$ Metric Calculation*. This makes D\&R computationally more efficient ($O(1)$ LLM call) and easier to deploy in black-box settings.
>
>
> #### **B2. Semantic Preservation (The "A loves B" Case)**
>
> **Response:** We thank the reviewer for pointing out this special case. We agree that in rare situations such as “A loves B” vs. “B loves A,” a permutation may produce a sequence that is not strictly semantically equivalent. Our claim of “semantic preservation” is meant at the token-level (i.e., preserving the semantic constituents) rather than guaranteeing full sentence-level meaning under all permutations. This does not affect our method, since the disrupted text is only an intermediate form and the recovery stage reconstructs the final coherent output.
>
> #### **B3. Pragmatic Similarity and Contextual Effects [2]**
>
> **Response:** We appreciate the reference to Ki et al. (*SEM 2024) [2] regarding the soundness of similarity metrics. We agree that embedding-based metrics like BERTScore effectively measure *pragmatic* similarity (contextual closeness) rather than strict *semantic* equivalence (e.g., “I love you” vs. “I don't hate you” might have high BERTScore).
>
> - **Mitigation Strategy:** This limitation of embedding metrics is precisely why D\&R incorporates **Structural Similarity metrics** (Kendall's $\tau$ and Spearman's $\rho$).
>   1. While “I love you” and “I don't hate you” might be pragmatically similar, their **structural word order** and token composition are distinct.
>
>   2. If an LLM recovers a pragmatically similar but syntactically distinct text, the **Structural Similarity** score will drop significantly, allowing the classifier to detect the deviation.

---

### Comment · Area_Chair_gYgy · 2025-11-21

Dear Reviewers,

We kindly encourage you to review and respond to the authors’ rebuttals. Your timely feedback is important for ensuring a fair and thorough review process. Thank you for your contributions to ICLR 2026.

AC

---

### Author Response · Authors · 2025-11-25
**Please feel free to share your feedback on our rebuttals ^^.**

Dear **Reviewers** and **AC**,

Again, thanks for your comments and your hard work! **Please feel free to share your feedback on our rebuttals ^^**.

Best,

Authors of submission 13074

---

### Author Response · Authors · 2025-11-29

**Dear Area Chair,**

We are writing to briefly summarize the contributions of our paper, "D&R: Recovery-Based AI-Generated Text Detection via a Single Black-Box LLM Call," and to provide context regarding our rebuttal process.

### Summary of Contributions
Our work addresses critical bottlenecks in AI-generated text detection, specifically the high computational cost of existing methods and their poor performance on short texts.

**State-of-the-Art Performance:** D&R achieves superior detection accuracy, attaining an AUROC of 0.96 on long texts and 0.87 on challenging short texts. It surpasses the strongest baselines by significant margins of +0.08 and +0.14, respectively.

**Theoretical Grounding:** Unlike heuristic approaches, our method is theoretically grounded in the "Concentration Assumption" and posterior concentration.

**Robustness & Generalization:** We demonstrate that D&R is robust to source-model mismatch and recovery-model variation, maintaining high performance even with smaller local models like Mistral-7B. Additional experiments in our rebuttal (RAID benchmark) confirm robustness against adversarial attacks and effectiveness in multilingual settings.

**Efficiency:** We introduce a single-call framework that drastically reduces computational overhead compared to multi-call baselines like RAIDAR and Fast-DetectGPT. Our method reduces latency from ~15s to 2s and cost from \\$5 to \\$0.2 per 1k samples.

### Rebuttal and Review Status
We received constructive and insightful feedback from four reviewers and we quickly submitted a comprehensive rebuttal that included the aforementioned additional experiments and clarified our theoretical presentation.

However, due to the recent unexpected circumstances affecting the review process, we regrettably did not receive final responses to our rebuttal from all reviewers. This is particularly unfortunate regarding **Reviewer H47X**, who explicitly stated that they would raise their score if their specific concerns were addressed.

We are confident that our rebuttal and the additional data provided in the revised manuscript have fully resolved these concerns. We kindly ask that you consider our detailed responses and the missed opportunity for a confirmed score adjustment when making your final recommendation.

Thank you for your time and service in handling this submission.

Sincerely,

Authors of submission 13074

---

### Meta-Review · Area_Chair_DU9o · 2026-01-05

**Summary:**

This paper proposed to use recovery-based (Within-Chunk Shuffling) approach to detect AI text. This intuition comes from the current LLM's window based pretraining strategy. Basically, the idea is novel (although the DNA-GPT is also one kind of recovery-based detection), this work emphasize "Within-Chunk Shuffling" which is consensus with the intuition of "pretraining" as the fingerprint is interesting and sound. I lean to accept.
THe only negative review is focusing on experiment part. The authors have addressed these concerns.

**Reviewer Concerns:**

The major concerns from reviewers are addressed.
For example, the only negative reviewer focus on experient part. In the rebuttal, the authors have included additional results to address them.
Some reviewers' comments are totally wrong, e.g.,
Ly38: "The method still trains a classifier on recovery features." This is a clear wrong comment. The work is zero-short approach.

**Reviewer Scores:**

Three reviewers give positive points, one with negative (4).
I read the rebuttal, and do think the issues are addressed.

---

### Decision · Program_Chairs · 2026-01-26

Accept (Poster)